# CONNECTING INDEPENDENTLY TRAINED MODES VIA LAYER-WISE CONNECTIVITY

## ABSTRACT

Empirical and theoretical studies have shown that continuous low-loss paths can be constructed between independently trained neural network models. This phenomenon, known as mode connectivity, refers to the existence of such paths between distinct modes-i.e., well-trained solutions in parameter space. However, existing empirical methods are primarily effective for older and relatively simple architectures such as basic CNNs, VGG, and ResNet, raising concerns about their applicability to more recent and structurally diverse models. In this work, we propose a new empirical algorithm for connecting independently trained modes that generalizes beyond traditional architectures and supports a broader range of networks, including MobileNet, ShuffleNet, EfficientNet, RegNet, Deep Layer Aggregation (DLA), and Compact Convolutional Transformers (CCT). In addition to broader applicability, the proposed method yields more consistent connectivity paths across independently trained mode pairs and supports connecting modes obtained with different training hyperparameters.

## 1 INTRODUCTION

The training of neural networks is often understood as the process of optimizing a non-convex loss function to locate local or global minima (Nguyen & Hein, 2017) in $\mathbb{R}^D$, where $D$ denotes the number of trainable parameters. An open question in Neural Network (NN) is what the low-loss region looks like within $\mathbb{R}^D$. Understanding the geometry and topology of the low-loss region may lead to better NN architectures and improve training efficiency. While we do not address this question directly, we develop an algorithm capable of finding low-loss paths between low-loss models, a.k.a. modes. We envision that this algorithm will allow researchers to navigate the low-loss region more systematically and gain insights into its geometric properties.

Early studies visualized the loss function, referred to as the *loss landscape*, suggesting that minima are points (Foret et al., 2021; Li et al., 2018a). However, these loss landscape visualizations are limited to low-dimensional projections, far smaller than the full parameter space, typically represented as $\mathbb{R}^D$. Subsequent empirical results indicate that two independently trained models with low loss can be connected by a continuous path along which all intermediate models also exhibit low loss (Garipov et al., 2018; Draxler et al., 2018; Fort & Jastrzebski, 2019; Benton et al., 2021). This surprising phenomenon is termed as *mode connectivity*, where a mode refers to a well-trained solution in $\mathbb{R}^D$.

Mode connectivity provides empirical evidence that the geometry of the loss landscape is more complex than a collection of isolated basins (Li et al., 2018a); instead, low-loss tunnels appear to connect these basins. It also offers insights into the shape of the low-loss region in $\mathbb{R}^D$, characterizing the set of parameters that achieve low training loss. Despite its potential, mode connectivity remains underexplored due to several limiting factors.

The primary limitation is the narrow range of model architectures supported by existing methods. Simple architectures require only simple tools—for example, in LeNet-5 on MNIST, linear interpolation between independently trained modes often produces a low-loss path (Frankle et al., 2020). In contrast, more complex architectures such as ResNet (He et al., 2015b), VGG (Simonyan & Zisserman, 2015), and DenseNet (Huang et al., 2018) on CIFAR-10 typically require specialized procedures such as AutoNEB (Draxler et al., 2018), which still do not consistently succeed across

different random seeds. Moreover, these architectures are now relatively dated, leaving an open question of whether mode connectivity persists in more recent and sophisticated models.

To address these limitations and to more efficiently navigate the low-loss region, we propose a new empirical algorithm, termed *Low-Loss Path Finding* (LLPF), for connecting two independently trained modes(i.e. trained with different random seeds). Our approach offers two main advantages: (1) it is effective not only for the aforementioned architectures but also for more recent models, including MobileNet (Sandler et al., 2019), ShuffleNet (Zhang et al., 2017), EfficientNet (Tan & Le, 2020), Deep Layer Aggregation (DLA) (Yu et al., 2019), RegNet (Radosavovic et al., 2020) and Compact Convolutional Transformers (CCT) (Hassani et al., 2022); and (2) it exhibits strong reproducibility—given appropriate search hyperparameters, it consistently discovers mode connections with nearly identical training loss and test accuracy trajectories irrespective of the random seed[1]. We attribute these improvements primarily to the use of *layer-wise mode connectivity*, which posits that two modes not linearly connected in the full parameter space may still be linearly connected in a layer-by-layer manner (Adilova et al., 2024).

**Contributions.** This paper makes the following contributions:

- We develop *Low-Loss Path Finding (LLPF)*, which constructs mode connections across a broader range of architectures than prior methods.
- LLPF exhibits high reproducibility: the discovered paths show consistent loss and accuracy trajectories across mode pairs trained with different random seeds (a property absent in prior work).
- LLPF could connect modes obtained with different training hyperparameters, a setting not evaluated by existing approaches.

**Paper Organization.** The remainder of this paper is organized as follows: Section 2 defines the terminology and concepts used throughout the paper. Section 3 reviews existing algorithms for connecting independently trained modes and compares them with our proposed approach. Section 4 presents the LLPF algorithm. Section 5 reports experimental results across a diverse set of architectures, including connections between modes trained under different hyperparameters. Section 6 discusses the limitations of LLPF. Finally, Section 8 concludes the paper.

## 2 TERMINOLOGY AND DEFINITIONS

To establish consistent terminology, we first revisit the procedure for training a neural network. The training process consists of the following steps:

1. Initialize model parameters $\theta_0 \in \mathbb{R}^D$ based on the model architecture $\mathcal{M}$.
2. Define a loss function $\mathcal{L}$ and an optimization function $\mathcal{O}$, where $\mathcal{O}$ includes all training hyperparameters $\xi$.
3. Prepare a dataset $\mathcal{D}$ and divide it into batches, denoted as $\mathcal{B}$, satisfying $\mathcal{B} \subseteq \mathcal{D}$.
4. Given model parameters $\theta_t$ and a batch $\mathcal{B}_t$, compute the loss $L_t = \mathcal{L}(\theta_t, \mathcal{B}_t)$.
5. Update model parameters using $\theta_{t+1} = \mathcal{O}(\theta_t, \nabla L_t)$, where $\nabla L_t$ represents the gradient of $L_t$ with respect to $\theta_t$.

Steps 4 and 5 together constitute a single training iteration, denoted as $\text{Train}(\theta_t, \xi, \mathcal{B}) \rightarrow \theta_{t+1}$, where $\xi$ encodes all training hyperparameters. For a well-designed model architecture $\mathcal{M}$ and an appropriate choice of $\mathcal{O}$, the loss $L_t$ is expected to converge to a local or global minimum, after sufficient optimization iterations. In this paper, we refer to any trained parameter $\theta$ that attains low training loss as a **mode**.

Since $\theta \in \mathbb{R}^D$, the parameters $\theta$ can be interpreted as a point $P$ in the high-dimensional parameter space $\mathbb{R}^D$[2]. Each point $P$ is also associated with a loss value $L$, which could be calculated with $L = \mathcal{L}(P, \mathcal{D})$. We define the low-loss space to be the collection of all points $P$ that result in a loss that is smaller than a threshold value $L_{\text{thres}}$. The low-loss space, denoted as $S_{L \leq L_{\text{thres}}}$, is defined in Equation 1.

$$S_{L \leq L_{\text{thres}}} \coloneqq \{P \in \mathbb{R}^D \mid \mathcal{L}(P, \mathcal{D}) \leq L_{\text{thres}}\} \quad \text{definition of } \textbf{Low-Loss Space} \tag{1}$$

---

[1]We validate this claim via hundreds of simulations; see Section 5.

[2]For geometric discussions we use $P$, and for machine learning context we use $\theta$ to indicate a model.

Because neural networks are structured in layers, the set of parameters $\theta$ can be decomposed as $[\theta_{l_0}, \ldots, \theta_{l_x}, \ldots \theta_{l_n}]$, where $l_x$ represents the layers defined in $\mathcal{M}$. Each layer's parameters are represented as $\theta_{l_x} \in \mathbb{R}^{d_{l_x}}$, where $d_{l_x}$ denotes the number of trainable parameters for layer $l_x$.

Since the layer parameters can be viewed as a distribution of real numbers, we define the variance of this distribution as $\mathrm{Var}(\theta_{l_x})$. We then define a region in $\mathbb{R}^{d_{l_x}}$ consisting of all points $\theta_{l_x}$ that satisfy the condition $\mathrm{Var}(\theta_{l_x}) = v$. This region, referred to as the **Variance Sphere** ($S_{var}$), forms a high-dimensional sphere in $\mathbb{R}^{d_{l_x}}$. For consistency, we use the point notation $P_{l_x}$ instead of $\theta_{l_x}$ in the formal definition of the Variance Sphere, given in Equation 2.

$$S_{var=v} := \{P_{l_x} \in \mathbb{R}^{d_{l_x}} \mid \mathrm{Var}(P_{l_x}) = v\} \quad \text{definition of } \textbf{Variance Sphere} \tag{2}$$

In typical model training procedures, the parameters of linear layers, convolutional layers, and transformer layers are initialized as a distribution with a specific variance and mean. The mean is usually set to zero, while the variance is determined by one of the following methods: LeCun (LeCun et al., 1998), Xavier (Glorot & Bengio, 2010), or Kaiming (He et al., 2015a). The number of trainable parameters also affects initialization, as all above methods compute variance based on the number of trainable parameters in the given layer. This relationship is mathematically expressed in Equation 3.

$$\theta_0, \theta'_0 = \mathrm{Init}(\mathcal{M}) \implies \mathrm{Var}(\theta_0) = \mathrm{Var}(\theta'_0) \tag{3}$$

Our experimental results (see Appendix A.1) and prior research (Chen et al., 2024) suggest that independently-trained modes (i.e., models initialized with different random seeds but trained with identical hyperparameters) tend to be positioned on variance spheres that are close to each other. This relationship is mathematically expressed in Equation 4.

$$\theta_n = \mathrm{Train}^n(\theta_0, \xi, \mathcal{D}), \quad \theta'_n = \mathrm{Train}^n(\theta'_0, \xi, \mathcal{D}) \implies \mathrm{Var}(\theta_n) \approx \mathrm{Var}(\theta'_n) \tag{4}$$

where train denotes a training iteration, $\xi$ represents all training hyperparameters (e.g., learning rate, momentum, weight decay) and $\mathcal{D}$ is the training dataset. The operator $\mathrm{Var}$ calculates the variance of model parameters $\theta$. Since a model consists of multiple layers, the result of the $\mathrm{Var}$ operator is a vector. For single-layer parameters $\theta_{l_x}$, the $\mathrm{Var}$ operator returns a scalar value.

In this paper, we make one approximation: the center of the variance spheres $S_{\mathrm{var}=v}$ is close to the origin, expressed as:

$$\mathrm{Mean}(\theta_{lx}) \approx 0 \tag{5}$$

This equation holds due to the following reasons: (1) during initialization, layer weights (excluding normalization layers) are set to a distribution with zero mean; (2) after training iterations, the deviation of the weights' mean is small and can be neglected. We justify this approximation empirically in Appendix A.2.

From Equation 5, we derive that the variance of the coordinates of a point $P$ (where $P \in S_{\mathrm{var}=v}$) is approximately proportional to the squared Euclidean distance from $P$ to the origin $O$ in $\mathbb{R}^{d_{l_x}}$, expressed as:

$$\|\overrightarrow{OP_{lx}}\|^2 \propto \mathrm{Var}(\theta_{lx}) \quad \text{(approximately)} \tag{6}$$

The full derivation is provided in Appendix A.10.

## 3 RELATED WORK AND COMPARISON

This section reviews existing work on algorithms for mode connectivity in independently trained models and compare them with our proposed LLPF algorithm. In addition, we discuss two related areas—mode connectivity in model spawning and model permutation—since they are sometimes mistakenly considered comparable to mode connectivity in independently trained models.

**Mode Connectivity in Independently Trained Modes**

The concept of mode connectivity was first introduced by Garipov et al. (2018), who proposed *Fast Geometric Ensembling* (FGE) to generate an ensemble model from two modes. Their method does not explicitly construct a mode connection, but rather states its existence. Shortly thereafter, Draxler et al. (2018) proposed *AutoNEB*, which incrementally bends a linear interpolation between two

modes until all intermediate points lie in the low-loss region. They also observed that the midpoint of the linear interpolation often forms a loss barrier, which AutoNEB attempts to overcome using stochastic gradient descent (SGD) (Robbins & Monro, 1951). Building on these insights, Benton et al. (2021) introduced *Simplicial Pointwise Random Optimization* (SPRO), which extends FGE by showing that the ensemble models forms a high-dimensional manifold.

While these works established foundational ideas, they also exhibit several limitations: (1) They were primarily tested on older architectures such as simple CNNs, ResNet, VGG, and DenseNet, raising concerns about their applicability to more recent architectures. (2) AutoNEB authors report that their method cannot guarantee finding a low-loss path for an arbitrary mode pair. Other methods only find individual points rather than a full path. (3) The connected modes in AutoNEB were obtained by using identical training hyperparameters. As suggested by Equation 4, such modes tend to lie on similar variance spheres, thereby limiting the generality of their conclusions.

Our proposed algorithm (LLPF) overcomes these limitations by extending applicability to more recent architectures, including MobileNet, EfficientNet, RegNet, ShuffleNet, DLA, and CCT. These models are selected based on their popularity and suitability for the CIFAR dataset, see Appendix A.11. LLPF provides consistent results across different random seeds and is capable of constructing reliable low-loss paths between modes located on different variance spheres. These statements are supported by the results in Section 5. A summary comparison with prior approaches is presented in Table 1.

Table 1: Comparison of existing mode connectivity algorithms and LLPF. "Tested model architectures" indicates the range of architectures evaluated. "Result consistency" indicates whether the algorithm reliably finds low-loss paths. "Different variance sphere" indicates whether the method has been tested on modes on different variance spheres, i.e., with different training hyperparameters. The final column reports the path's worst-case training loss on CIFAR10(maximum training loss along the path); lower is better.

| Method | Tested model architectures | Result consistency | Different variance sphere | Worst case training loss |
|---|---|---|---|---|
| AutoNEB (Draxler et al., 2018) | Basic CNN, ResNet, DenseNet | Inconsistent | N/R | 0.0324 (ResNet-20) |
| FGE (Garipov et al., 2018) | ResNet, VGG, WideResNet | N/R | N/R | 0.022 (ResNet-158) |
| SPRO (Benton et al., 2021) | VGG, ResNet | N/R | N/R | N/R |
| LLPF (our) | All above and EfficientNet, MobileNet, RegNet, ShuffleNet, DLA, CCT | Consistent | Tested to support | 0.006 (ResNet-18) |

*Notes:* N/R = not reported. Note that FGE and SPRO are model-ensembling methods that identify only single points on the mode connection rather than constructing a full continuous path. Worst-case training losses are not strictly comparable across papers due to differing models.

**Linear Mode Connectivity in Spawning and Permutation**

Linear Mode Connectivity (LMC) is a closely related area investigating the observation that, in certain cases, two modes can be connected through simple linear interpolation. For relatively simple models and datasets, such as LeNet on MNIST, linear interpolation often yields a low-loss path Frankle et al. (2020). For more complex architectures, however, the difficulty of constructing such connections depends strongly on how the modes are obtained. Nagarajan & Kolter (2021); Frankle et al. (2020); Zhou et al. (2023); Juneja et al. (2023) have shown that linear interpolation often suffices when the modes are related through one of the following processes:

**Model Spawning** (Frankle et al., 2020; Fort et al., 2020): a model is randomly initialized and trained for a few epochs, then spawned into two copies and continue to be independently trained until convergence.

**Model Permutation** (Ainsworth et al., 2023; Entezari et al., 2022): two models are trained independently, after which the neurons of one are permuted to align with those of the other, yielding a functionally equivalent representation.

A common pitfall in the literature is conflating these settings with true independent training. For example, Garipov et al. (2018) reported mode connectivity between two ResNet-50 models on ImageNet-1k, but both were fine-tuned from the same pretrained model with different hyperparameters, which constitutes model spawning rather than independent training.

In this paper, *we explicitly restrict our scope to mode connectivity between independently-trained models without further permutation*, i.e., models initialized with different random seeds and trained to convergence.

## 4 LOW-LOSS PATH FINDING ALGORITHM

We propose the *Low-Loss Path Finding* (LLPF) algorithm to connect two independently trained modes. LLPF is composed of two complementary components. The first, referred to as the *model-to-model* algorithm (LLPF_M2M), constructs a path between two modes located on similar variance spheres—that is, satisfying the condition in Equation 4. This procedure is detailed in Algorithm 1. The second component, the *model-to-origin* algorithm (LLPF_M2O), generates a path from a low-loss model toward the origin of $\mathbb{R}^D$, as described in Algorithm 2. These components serve distinct purposes: Algorithm 1 enables exploration within the same variance sphere, while Algorithm 2 enables exploration across variance spheres.

The geometric intuition behind Algorithm 1 is illustrated in Figure 1, using the notation of the algorithm, including points $P_0$, $P_1$, $M_1$, $M_2$, and $M_3$. The algorithm starts by moving the starting $P_0$ slightly toward the destination $D$ to obtain $M_1$ via a weighted average(Move, Line 5), controlled by hyperparameters step_f, step_a and step_c. Next, the VarianceCorrection step projects $M_1$ back as $M_2$ onto the variance sphere of $P_0$ to counteract the vanishing variance problem, a phenomenon wherein averaging uncorrelated neural networks leads to reduced parameter variance, thereby hindering subsequent training efforts (Tian et al., 2024). $M_2$ is refined through $r$ training steps to reduce loss below threshold, yielding $M_3$, which is again projected back to the sphere to produce $P_1$, which serves as the new starting point for the next iteration. Repeating this process for $T$ iterations yields a low-loss path $\{P_i\}$ from $P_0$ to $P_T \approx D$.

The mechanism of Algorithm 2 is similar to that of Algorithm 1, with two key differences: the removal of the *VarianceCorrection* step and the inclusion of an *AngleConformal* step. The *AngleConformal* step is introduced to adaptively reduce the learning rate as the variance decreases. This adjustment is necessary because applying the learning rate appropriate for a large variance sphere[3] to a small variance

**Algorithm 1** LLPF_M2M: Algorithm for constructing a low-loss path between two models. The major steps of this algorithm are also illustrated in the geometry plot in Figure 1, which uses consistent point notations as this algorithm.

> **function** LLPF_M2M($P_0, D$)     ▷ $P_0$: starting point, $D$: destination point
2:     $P_0, D \in$ *Flat Low Loss Region*
    $P_i = P_0$
4:     **for** $i = 0$ to $T$ **do**
       $M_1 = \text{Move}(P_i, D, step_f, step_a, step_c)$    ▷ Corresponding to $\overrightarrow{P_0 M_1}$
6:        $M_2 = \text{VarianceCorrection}(M_1, S_{var})$ ▷ Corresponding to $\overrightarrow{M_1 M_2}$
       $M_3 = \text{Train}^r(M_2, \xi, \mathcal{B})$    ▷ Corresponding to $\overrightarrow{M_2 M_3}$. Training $r$ steps with hyper-parameters $\xi$
8:        $P_{i+1} = \text{VarianceCorrection}(M_3, S_{var})$    ▷ Corresponding to $\overrightarrow{M_3 P_1}$
    **end for**
10:     **return** $P_0 \ldots P_n$     ▷ Final output of LLPF_M2M
    **end function**
12:
    **function** MOVE($P_i, D, step_f, step_a, step_c$)
14:     $step = step_a |\overrightarrow{P_i D}| + step_c |\overarc{P_0 D}| + step_f$
    $M_1 = P_i + step \overrightarrow{P_i D}$
16:     **return** $M_1$
    **end function**
18:
    **function** VARIANCECORRECTION($W, S_{var=v}$)     ▷ $W$ is treated as an array
20:     $\bar{W} = \frac{1}{n} \sum_{i=1}^{n} W[i]$   ▷ Compute mean of $W$, $i$ indicates the element index
    $\sigma_W^2 = \frac{1}{n} \sum_{i=1}^{n} (W[i] - \bar{W})^2$ ▷ Compute variance of $W$
22:     **for** $i$ in len($W$) **do**    ▷ $i$ indicates the element index
       $W'[i] = \bar{W} + \sqrt{\frac{v}{\sigma_W^2}}(W[i] - \bar{W})$   ▷ Scale variance
24:     **end for**
    **return** $W'$
26: **end function**

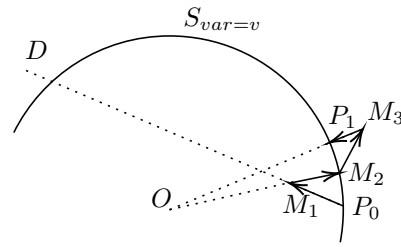

Figure 1: Geometric illustration of the first iteration of Algorithm 1, showing the transition from $P_0$ to $P_1$. Point labels follow the algorithm's notation. The origin is used as the center of the variance sphere, assuming the mean is approximately zero (Equation 5 and Equation 6).

---

[3]A large variance sphere refers to a variance sphere with a high variance value.

sphere often causes the model to deviate from the low-loss path. Although Algorithm 2 can theoretically be applied in reverse (to move a model toward a larger variance sphere) we observe that this process frequently suffers from exploding gradients problem (Hanin, 2018). Consequently, the reverse direction is omitted.

**Hyperparameters**

In addition to traditional hyperparameters such as learning rate and optimizer, there are several algorithm-specific hyperparameters and design principles to consider, which we discuss below.

The most critical hyperparameters for successfully finding a low-loss path using Algorithm 1 are the *choice and order of layers*. This is because the method operates one layer at a time, as the variance sphere is defined per layer (see $d_{l_x}$ in Eq. 2). Applying it to all layers simultaneously works for simple models (e.g., LeNet5@MNIST) but fails for more complex ones (e.g., ResNet18@CIFAR10). In these cases, layer-wise mode connectivity (Adilova et al., 2024) provides a suitable framework, and Algorithm 1 should be applied sequentially in a carefully determined order. We find that an empirically effective ordering strategy, which follows two principles: (1) layers should be processed in the direction of data flow, typically from shallow to deep until the final output layer; and (2) parallel layers, such as attention modules following image patching, should be processed individually in any order before proceeding to subsequent layers. We refer to this approach as the Follow Data Flow (FDF) strategy, and illustrations of applying FDF on DLA and CCT7 are provided in Appendix A.5, Table 2.

> **Algorithm 2** Algorithm to find the low-loss path across different $S_{var}$.
>
> **function** LLPF_M2O($P_0$)      ▷ $P_0$: starting point
> 2:    $P_0 \in$ *Flat Low Loss Region*
>      $P_i = P_0$
> 4:    $P_0 \in S_{var=v}$      ▷ Variance of $P_0$ is $v$
>      **for** $i = 0$ to $n$ **do**
> 6:        $N = $ Move($P_i, O, step_f, step_a, step_c$)    ▷ Move toward origin $O$
>        $\xi = $ AngleConformal($N, v$)      ▷ Adjust training hyper-parameters based on $N$'s variance
> 8:        $P_{i+1} = $ Train$^r$($N, \xi, \mathcal{B}$)      ▷ Training $r$ steps with hyper-parameters $\xi$
>      **end for**
> 10: **end function**
>
> 12: **function** ANGLECONFORMAL($N, v$)
>      $N \in S_{var=w}$      ▷ calculate the variance of $N$ to be $w$
> 14:    $\eta = \frac{\eta_{base} \cdot \overline{w}}{v}$ ▷ Scale learning rate $\eta$ to maintain angular conformity
>      $\xi = \eta, \ldots$      ▷ Include all other hyper-parameters
> 16:    **return** $\xi$
>      **end function**

For Algorithm 1, the optimizer should exclude regularization techniques such as weight decay, as the constraint of remaining within a single variance sphere already ensures that the variance of the layer weights is preserved. Additionally, momentum is generally avoided in this setting. While momentum is usually used to stabilize long-distance training trajectories, it is less suitable here, as the update steps in Algorithm 1 are small and localized.

Although Algorithm 1 exposes many hyperparameters which can complicate its application, in practice only the *layer selection and order* determine success. Most remaining hyperparameters affect path quality rather than feasibility. For example, $step_a, step_c$, and $step_f$ control path continuity—smaller values lead to smaller distances between consecutive points $P_i$ and $P_{i+1}$. The parameter $r$ determines how low the training loss can go—larger values correspond to more training iterations and potentially lower loss. However, high-quality paths [4] also come at the cost of increased computational workload, so the computation budget must be considered.

Algorithm 2 shares similar hyperparameters with Algorithm 1, but with two key differences: (1) the selection and order of layers is less critical, as moving all layers simultaneously successfully finds a low-loss path in our experiments; (2) normalization layers should be excluded from Algorithm 2, since their purpose is to rescale outputs to zero mean and unit variance, and moving them toward the origin reduces their weight variance, thereby impairing their functionality.

**Prerequisite**

Algorithm 1 has one key prerequisite: the corresponding layers of the input models must lie on nearby variance spheres, as defined in Equation 4. We also recommend selecting training hyperparameters that achieve near-zero training loss, as such solutions tend to exhibit high predictive confidence, which is associated with flatter regions of the loss landscape Walter et al. (2025). This helps reduce the probability that the path becomes trapped in local minima. In Appendix A.9, we additionally provide a ResNet18@CIFAR10 example trained with a constantly large learning rate

---

[4]Low-loss paths with low maximum loss and high resolution.

and zero weight decay. This experiment shows that Algorithm 1 can still succeed even when the modes are located in sharp minima and obtained under poorly chosen training hyperparameters.

## 5 EXPERIMENTAL EVALUATION OF LOW-LOSS PATH FINDING ALGORITHMS

This section presents the results of applying Algorithm 1 and Algorithm 2 to construct low-loss paths between two independently trained modes across the architectures discussed in Section 1. For ResNet18@CIFAR10, DLA@CIFAR10, and CCT7@CIFAR10, the results are shown in Figure 2 and Figure 3. To ensure consistency of the results, we repeat each experiment at least 10 times, using independently trained low-loss models with identical hyperparameters[5]. These repetitions demonstrate that our algorithm consistently produces low-loss paths across different mode pairs, as summarized in Table 1. The full hyperparameter configurations are provided in Appendix A.5.

For other architectures (including those mentioned earlier but not detailed here) and CIFAR100 dataset, the results of Algorithm 1 and Algorihtm 2 are presented in Appendix A.3 and Appendix A.4, respectively.

In Figure 2, we assess path validity using three criteria: (1) the training loss and accuracy should stay in the region that is considered as low-loss and high-accuracy (first row); (2) the distance between $P_i$ and $D$ should gradually converge to zero (second row), with minor deviations allowed due to the approximation in Equation 4; and (3) the final testing loss should converge to the testing loss at iteration 0, since the final point is expected to converge to $D$, which should exhibit similar generalization behavior as $P_0$. In our results, all training losses remain below 0.1, and the final layer-wise distances gradually converges below $10^{-1}$, indicating that the low-loss paths in Figure 2 are correctly constructed.

The third row of Figure 2 shows test accuracy and loss along the paths. However, we observe that Algorithm 1 does not guarantee low testing loss, as seen in the ResNet18 case, where test loss increases despite low training loss. This is because the algorithm operates within the training-defined low-loss space (Equation 1), which is not align with the testing dataset.

Figure 3 demonstrates that Algorithm 2 can connect modes across different variance spheres. This is evident in several aspects: (1) the training loss remains low throughout the path; and (2) the $L_2$ distance decreases gradually as the path crosses different variance spheres, since the $L_2$ distance measures the distance from the current point to the origin (Equation 6). Unlike the results in Figure 2, the $L_2$ distance here is not expected to converge to zero, because the destination point in Algorithm 2 is the origin, which is not itself a low-loss mode. Nevertheless, using the ResNet18@CIFAR10 case, we empirically show that Algorithm 2 can visit most of the variance spheres on which SGD-trained modes are located, see Appendix A.8.

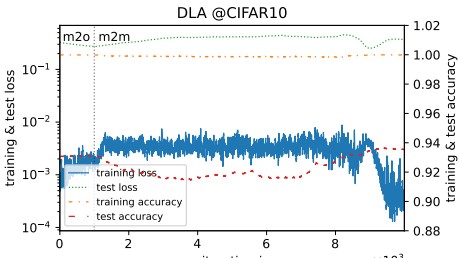

Figure 4: Connecting two modes on different variance spheres. Iterations 0–1000 ("m2o") use Algorithm 2 to reach the target variance sphere; the subsequent "m2m" phase uses Algorithm 1 to reach the destination mode. Training accuracy remains high throughout, indicating a valid connection. Final test accuracy differs from the initial value due to weight-decay–induced generalization changes.

**Connecting Modes on Different Variance Sphere**

We validate the claim that LLPF supports connecting modes located on different variance spheres, as mentioned in Table 1, using the DLA@CIFAR10 case. Specifically, we trained two modes with different weight decay and learning rate parameters [6], yielding $P \in S_{var=v_0}$ and $D \in S_{var=v_1}$, where $D$ is trained with a larger weight decay than $P$. Since stronger weight decay tends to position models closer to the origin (Li et al., 2018b), we obtain the relationship $v_0 > v_1$.

Constructing connectivity across different variance spheres requires two steps: (1) apply Algorithm 2 to find a low-loss path from point $P$ to an intermediate point $I$ on $S_{var=v_1}$; and (2) apply

---

[5]Training-accuracy curves are shown for a single representative run.

[6]Hyperparameters are available in Table 4.

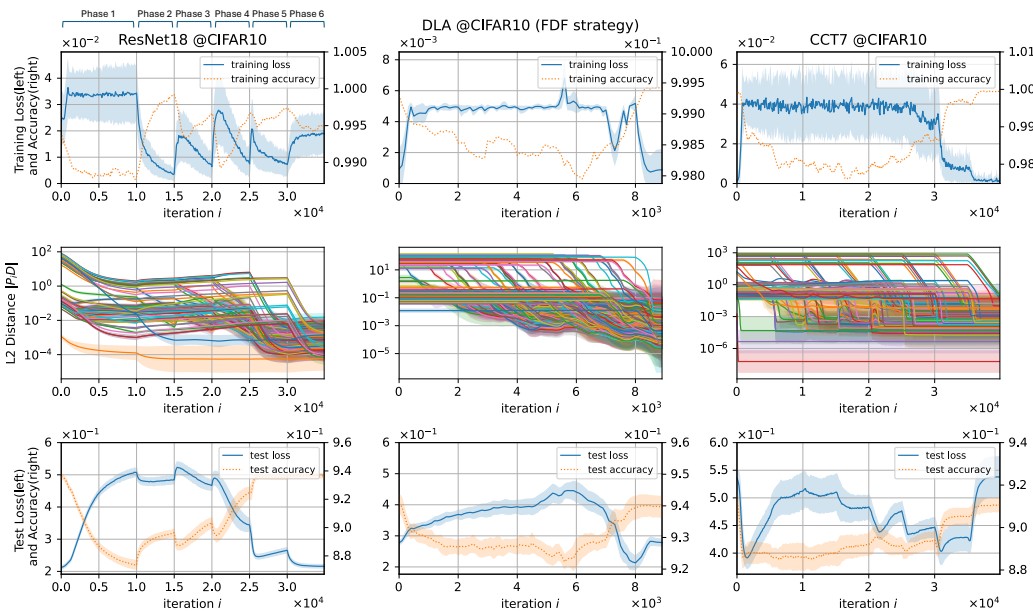

Figure 2: Results of Algorithm 1. Additional results on CIFAR100 dataset and other model architectures are available in Appendix A.3. Each column corresponds to a different model and dataset: ResNet18@CIFAR10 (first column), DLA@CIFAR10 (second column), and CCT7@CIFAR10 (third column). The first row shows the training loss averaged across the final training rounds(near $M_3$) and the training accuracy of one experiment; the second row shows the layer-wise $L_2$ distance of $P_iD$; and the third row shows the test loss and test accuracy at $P_i$ of the whole testing dataset. Legends for the second row are omitted due to the large number of layers. Each experiment is repeated with different pairs of starting and destination modes, trained from different random seeds. The curves show the mean across repetitions, and the shaded regions indicate the standard deviation. Both the curves and the shaded boundaries are smoothed using a moving average with a window size of 10. The distinct patterns observed in the training loss and layer-wise $L_2$ distance reflect which layers are moved in each phase, see Table 2. For ResNet18, the duration of each phase is marked above the training-loss panel. Full phase configurations are listed in Table 2.

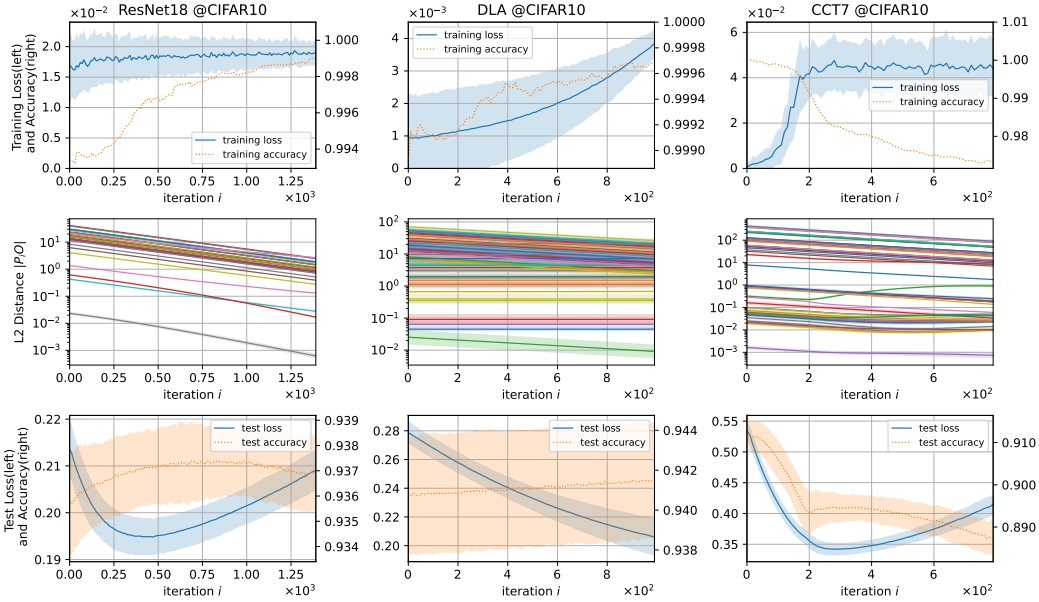

Figure 3: Results of Algorithm 2, following the same layout and postprocessing steps as in Figure 2, except that all normalization layers are excluded from the second row, as Algorithm 2 does not operate on them. Since the origin is not a low-loss model, the training loss is not expected to remain low, nor is the layer-wise $L_2$ distance expected to reach zero.

Algorithm 1 to connect $I$ with $D$. The full connectivity path is then formed by concatenating the results of steps (1) and (2).

The hyperparameters for both steps are nearly identical to those used in Figures 2 and 3, except that the destination point of Algorithm 2 is not the origin but the projection of $P$ onto $S_{var=v_1}$. The results for the DLA@CIFAR10 case are shown in Figure 4, and additional experiments on ResNet18@CIFAR10 and CCT7@CIFAR10 are provided in Appendix A.6.

**Mode-Connection Continuity Check**

The outputs of Algorithm 1 and Algorithm 2 are sequences of discrete points obtained via SGD, which raises concerns about the continuity of the constructed paths. We posit that the linear interpolation between consecutive points $P_t$ and $P_{t+1}$ also remains within the low-loss region. This argument is supported by prior theoretical work (Neyshabur et al., 2018), which shows that small perturbations to model weights typically do not substantially affect a network's outputs. Complementing this theoretical basis, we empirically validate continuity by linearly interpolating between $P_t$ and $P_{t+1}$ and measuring the training loss along the interpolation in the CCT7@CIFAR10 case. The results in Figure 5 confirm that these interpolated segments maintain low loss, thereby supporting the continuity of LLPF-generated connections. Appendix A.7 provides additional checks for DLA@CIFAR10 and ResNet18@CIFAR100.

**Minimization of Training Loss Tolerance**

A natural question is how to set the threshold that defines *low loss*. In standard training on CIFAR10 dataset, the loss can often be driven arbitrarily close to zero with sufficient iterations; path construction differs because each intermediate point is optimized under a limited compute budget and small, local updates.

We define the *training-loss tolerance* as the gap between the training loss of the terminal modes and

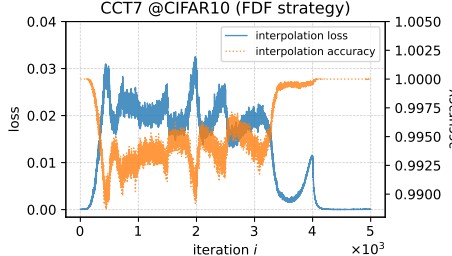

Figure 5: Training-set loss along linear interpolations between $P_t$ and $P_{t+1}$; 50 interpolation samples per segment. The training loss curve differs from Figure 2 because we slightly adjusted hyperparameters and adopt FDF strategy to reduce the computational cost of evaluating interpolation points.

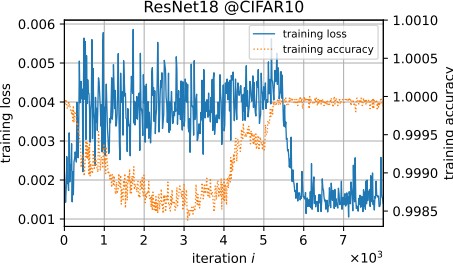

Figure 6: Training loss along a mode-connection path for ResNet18@CIFAR10. Hyperparameters are tuned so that the maximum training loss along the path stays below 0.006.

the largest training loss observed along the connection path. We empirically examine the minimum tolerance achievable by LLPF. With targeted hyperparameter tuning and additional compute, we obtain a ResNet18@CIFAR10 path whose maximum training loss remains below 0.006 (Figure 6), substantially tighter than the 0.04 reported in Figure 2. This indicates that a smaller loss tolerance is attainable at the cost of extra hyperparameter tuning and increased computation.

## 6 LIMITATIONS

The major limitation of our approach is that it does not guarantee *model generalization ability* along the discovered low-loss paths. Here, generalization refers to performance on unseen data, which we operationalize as test accuracy. Empirically, methods that construct paths by starting from a linear interpolation and explicitly overcoming the associated loss barrier (e.g., AutoNEB) often identify low-loss trajectories that also preserve favorable test performance (Draxler et al., 2018). By contrast, our algorithm is confined to training-defined low-loss regions and can therefore exhibit discrepancies between training loss and test metrics along the path.

## 7 DISCUSSION

The main contribution of this paper is the LLPF algorithm, which enables reliably finding low-loss paths between independently trained SGD modes obtained from different random seeds. Beyond this algorithmic contribution, the empirical findings presented here also raise several interesting points worthy of further discussion.

For the model architectures and dataset pairs studied in this work, the following observations apply.

**Loss landscape, basin, flat directions and low-loss tunnels.** Existing visualizations of loss landscapes usually suggest that they resemble isolated basins. Subsequent studies have shown that these basins possess flat directions along which one can move while maintaining near-zero loss (Wei et al., 2023; Geiger et al., 2019). Our empirical results suggest that the structure is even more complicated: rather than merely containing local flat manifolds, the loss landscape also contains low-loss tunnels that connect independent minima.

**Full path-connectedness of SGD basins** The narrow shaded region in Figure 2 and Figure 3 indicate that the resulting low-loss paths exhibit highly consistent properties, including similar training and test performance. This suggests a plausible hypothesis: for independently trained modes obtained via SGD, it may always be possible to construct a low-loss path connecting them, regardless of random seeds or training hyperparameters. If true, this would imply that all such modes lie within a single path-connected low-loss manifold of parameter space, in the sense that one can traverse it continuously without leaving the low-loss region.

However, to the best of our knowledge, no existing theoretical framework formally supports this empirical conjecture. Current theoretical explanations address mode connectivity only on the same variance sphere and remain limited to relatively simple CNN and VGG architectures, while relying on strong assumptions (Kuditipudi et al., 2020; Nguyen et al., 2021; Lubana et al., 2023).

## 8 CONCLUSION

We introduced *Low-Loss Path Finding* (LLPF), a practical algorithmic framework for constructing low-loss connections between independently trained modes. LLPF leverages *layer-wise mode connectivity* to operate effectively across diverse architectures, including ResNet, DenseNet, VGG, MobileNet, EfficientNet, RegNet, ShuffleNet, DLA, and CCT. From our experimental results on CIFAR10, LLPF offers two key advantages compared with prior approaches: (1) **reproducibility and consistency**—for modes obtained with different random seeds, it consistently discovers connections with nearly identical training/test loss and accuracy trajectories. (2) **cross-hyperparameter applicability**—it support connecting modes trained under different hyperparameters (i.e., modes on different variance spheres), a setting not evaluated in earlier work. Additionally, with appropriate search hyperparameters and sufficient compute, LLPF attains a lower worst-case (maximum) training loss along the path than existing methods (e.g., 0.006 on ResNet18@CIFAR-10).

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

## A  APPENDIX

### A.1  EMPIRICAL VALIDATION OF EQUATION 4

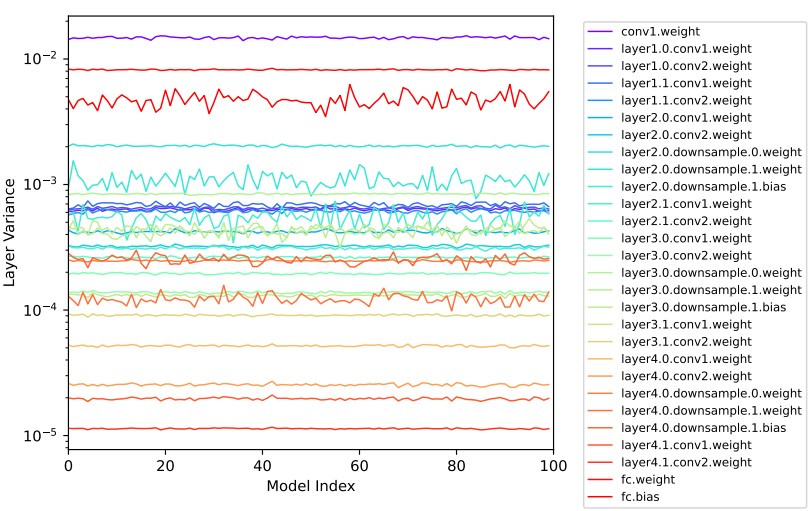

Figure 7: Layer-wise weights variance across 100 independently trained ResNet-18 models on the CIFAR-10 dataset, each initialized with a different random seed. The x-axis denotes the model index, and the y-axis represents the variance of each layer's parameters on a logarithmic scale. Each curve corresponds to one layer, following the PyTorch naming convention. The results show that layer variances remain consistent across different runs. Batch normalization layers are excluded, as their statistics are highly sensitive to the training batches.

### A.2  EMPIRICAL VALIDATION OF EQUATION 5

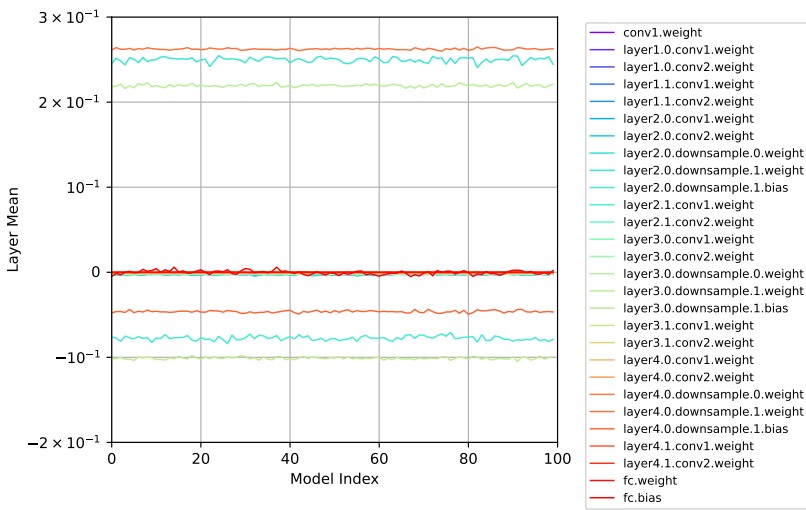

Figure 8: Layer-wise weights mean across 100 independently trained ResNet-18 models on the CIFAR-10 dataset. The figure layout follows that of Figure 7. The results indicate that the mean values of most layers are close to zero.

## A.3 ADDITIONAL RESULTS FOR ALGORITHM 1

In this section, we present the results of applying Algorithm 1 to additional model architectures in Figure 9, and to the CIFAR100 dataset in Figure 10.

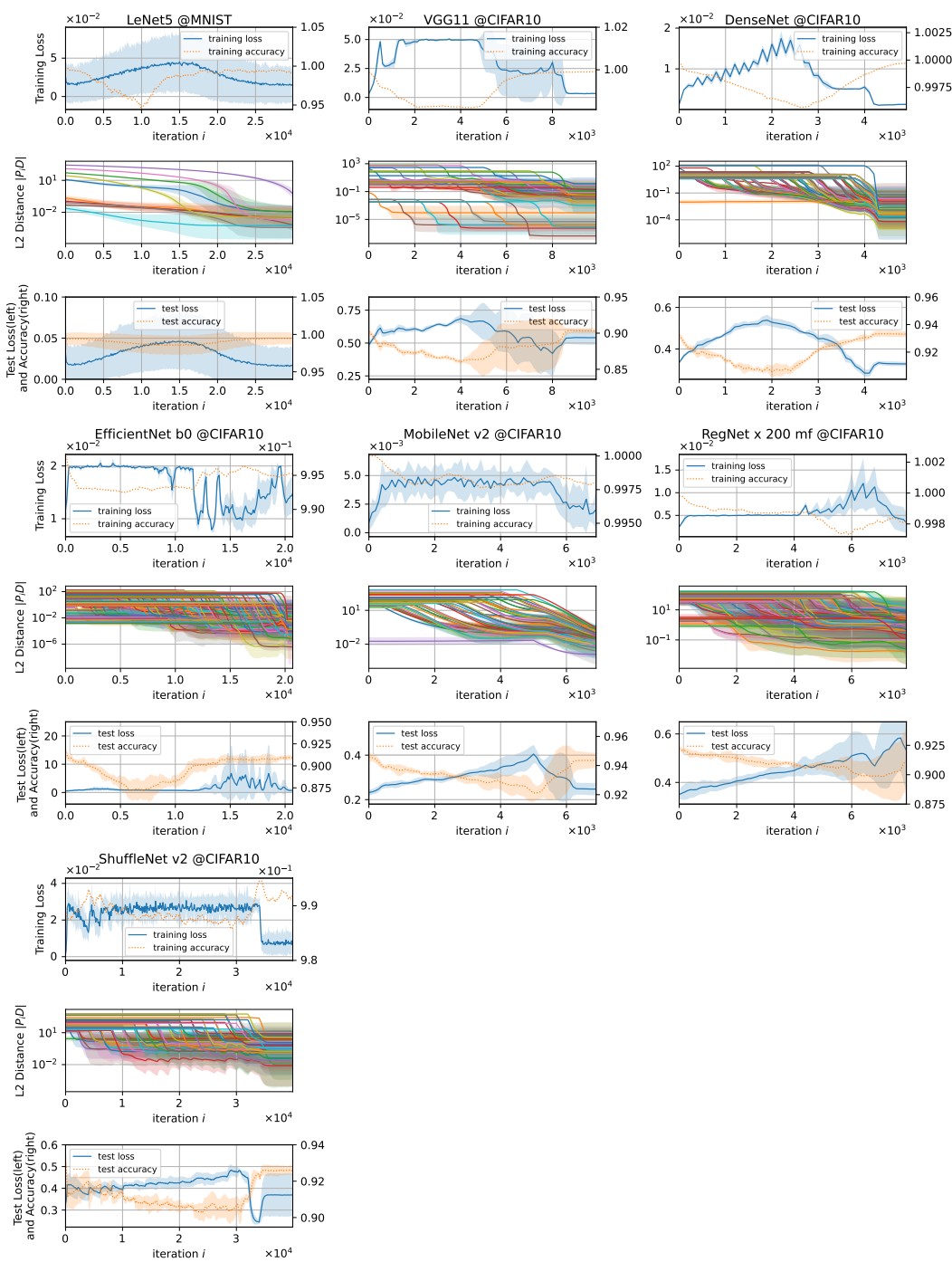

Figure 9: Additional results for Algorithm 1 across diverse architectures: LeNet-5, VGG-11, DenseNet, EfficientNet-B0, MobileNet-V2, RegNetX-200MF, and ShuffleNet. This figure extends Figure 2 and follows the same layout. For RegNet@CIFAR-10, test accuracy/loss exhibit higher variance at final stages because the two endpoint modes do not fully satisfy Equation 4, with up to a $3\times$ difference in certain layers.

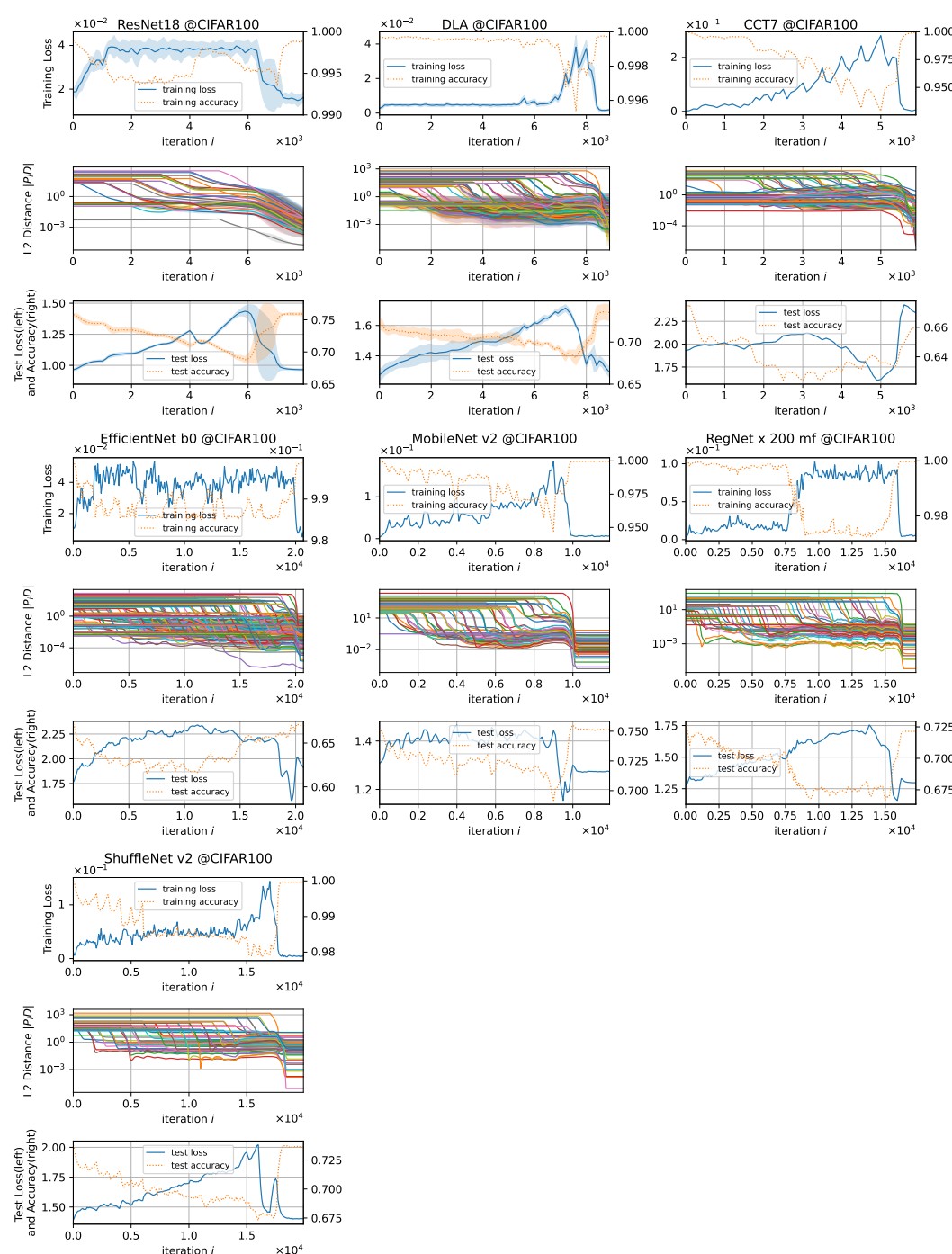

Figure 10: Additional results of Algorithm 1 on the CIFAR100 dataset. For the ResNet18@CIFAR100 and DLA@CIFAR100 experiments, each setting is repeated three times with different random seeds, and the shaded regions behind the curves indicate the standard deviation across these repetitions. All other experiments are conducted only once.

A.4    ADDITIONAL RESULTS FOR ALGORITHM 2

In this section, we present the results of applying Algorithm 2 to additional model architectures in Figure 11, and to the CIFAR100 dataset in Figure 12.

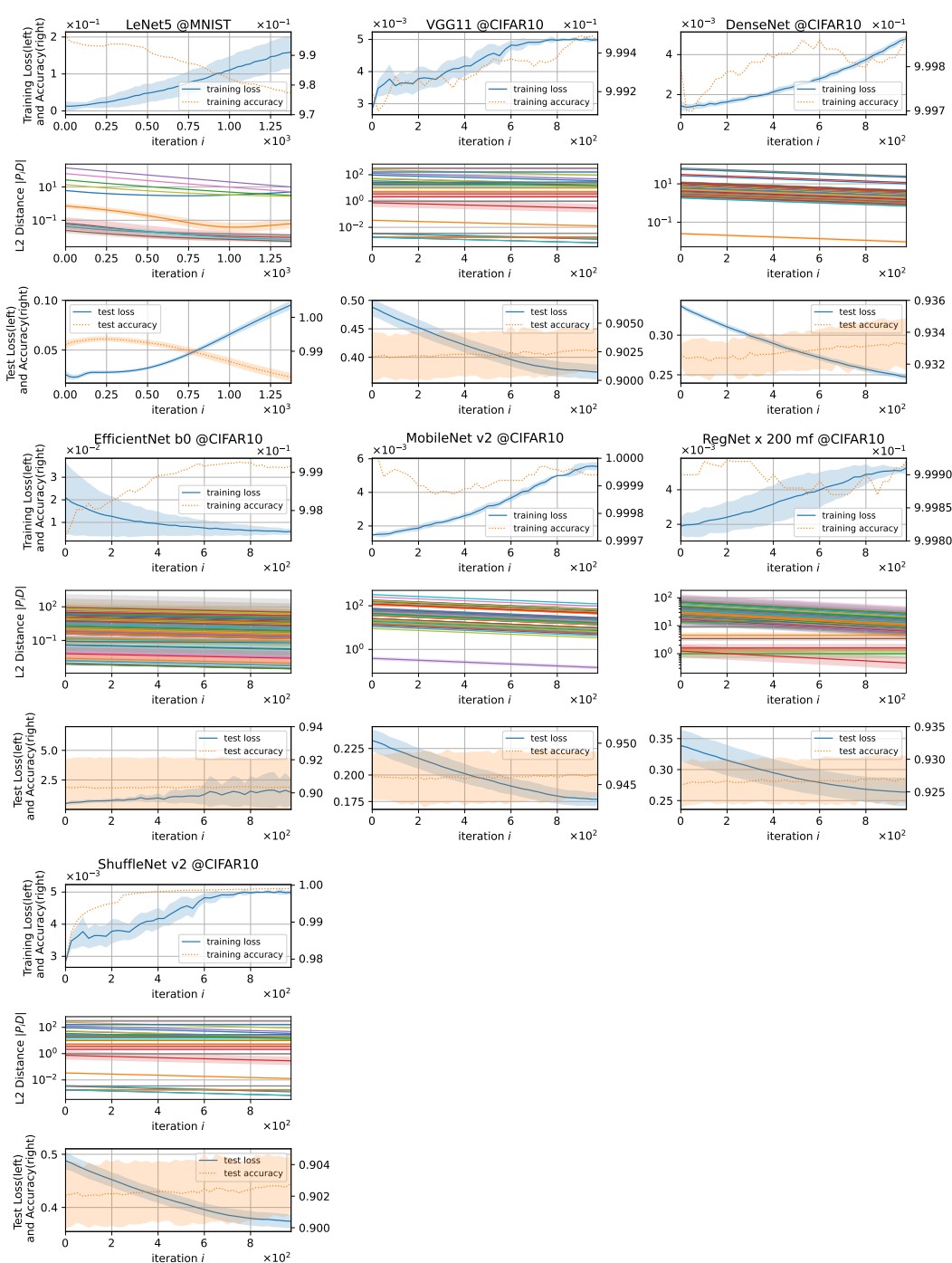

Figure 11: Additional results for Algorithm 2 across the corresponding architectures in Figure A.4. Notably, training accuracy drops markedly for LeNet5@MNIST because LeNet-5 lacks normalization layers; moving the model toward the origin reduces layer-weight variance, which in turn lowers output variance and degrades accuracy.

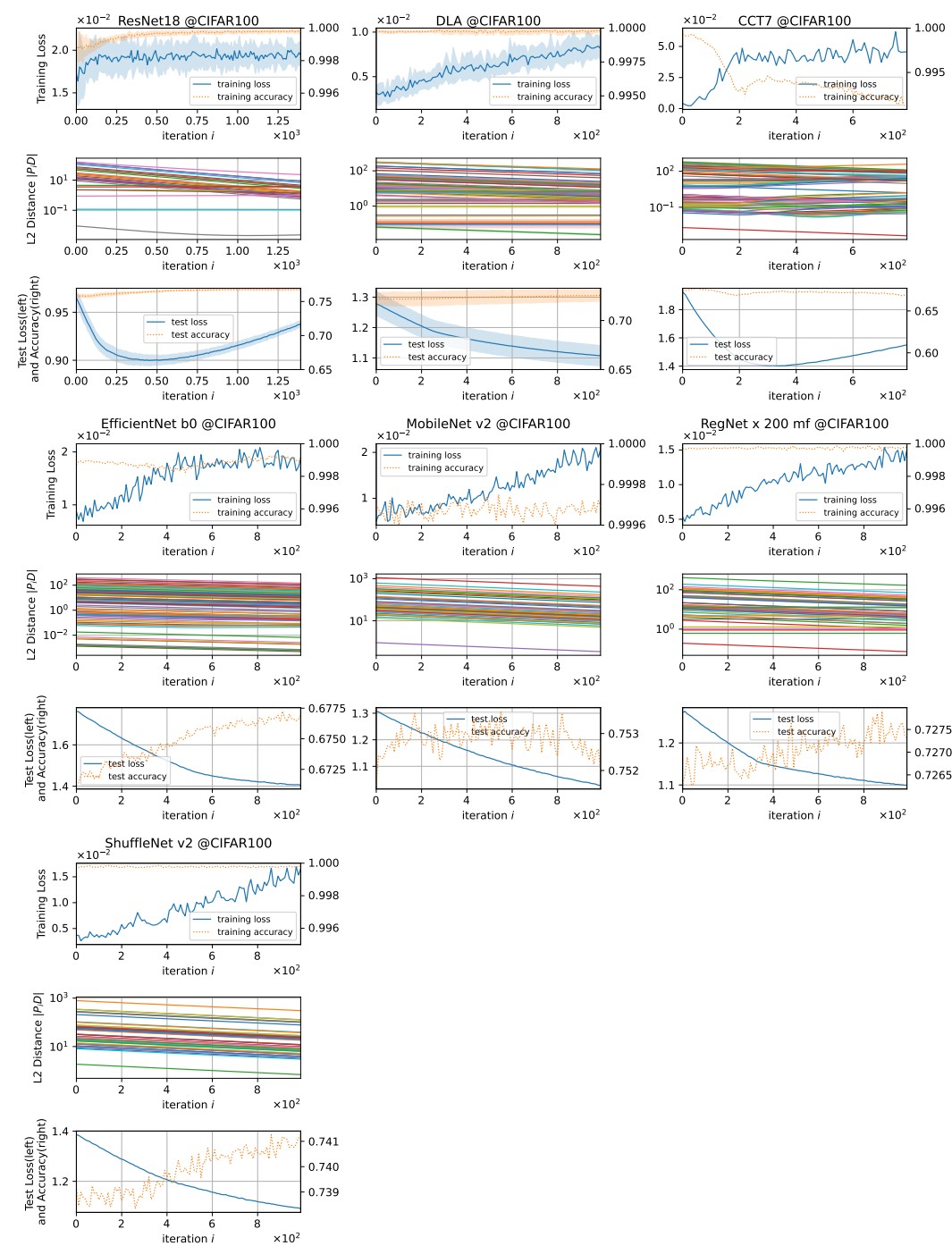

Figure 12: Additional results of Algorithm 2 on the CIFAR100 dataset. The ResNet18@CIFAR100 and DLA@CIFAR100 experiments are repeated three times, following the setup in Figure 10.

## A.5 HYPERPARAMETER TABLE AND OTHER CONFIGURATIONS

Table 2: Hyperparameters used for applying Algorithm 1 to connect modes for LeNet5 @MNIST, ResNet18 @CIFAR10, CCT7 @CIFAR10 and DLA @CIFAR10. For ResNet18, CCT7 and DLA, the exploration is conducted in multiple phases, each involving only a subset of layers, as specified in the "Applied Layers" column. Models not listed in this table employ the FDF strategy to determine the layer order. Layer names follow the PyTorch implementation of the corresponding models. Hyperparameters for other models is available in the supplementary material. Unmentioned $step$ hyperparameters are set to zero.

| | Applied layers | Iteration T (A1 L4[a]) | Train round $r$ (A1 L7) | Batch size $\mathcal{B}$ (A1 L7) | Optimizer and hyperparameter x (A1 L7) | step_a (A1 L14) | step_f (A1 L14) |
|---|---|---|---|---|---|---|---|
| **LeNet5 @MNIST** | | | | | | | |
| Phase 1 | all layers | 30000 | 5 | 64 | SGD($\eta$=0.001, $\beta$=0) | 0 | 1e-3 |
| **ResNet18 @CIFAR10** | | | | | | | |
| Phase 1 | all layers | 10000 | | | | 5e-4 | 0 |
| Phase 2 | conv1.weight bn1.{weigt+bias} fc.{weight+bias} | 5000 | train until loss[b]<0.04 or $r$>200 | 256 | SGD($\eta$=0.001, $\beta$=0) | 1e-3 | 0 |
| Phase 3 | Phase 2[c] + layer1.*[d] | 5000 | | | | 1e-3 | 0 |
| Phase 4 | Phase 3 + layer2.* | 5000 | | | | 1e-3 | 0 |
| Phase 5 | Phase 4 + layer3.* | 5000 | | | | 1e-3 | 0 |
| Phase 6 | all layers | 5000 | | | | 1e-3 | 0 |
| **CCT7_3x1_32 @CIFAR10** | | | | | | | |
| Phase 1 | tokenizer.conv_layers+ classifier.* exclude blocks | 5000 | | | | 1e-3 | 1e-3 |
| Phase 2 | Phase 1 + classifier.blocks.0.* | 5000 | | | | 1e-3 | 1e-3 |
| Phase 3 | Phase 2 + classifier.blocks.1.* | 5000 | train until loss<0.05 or $r$>10000 | 128 | SGD($\eta$=0.001, $\beta$=0) | 1e-3 | 1e-3 |
| Phase 4 | Phase 3 + classifier.blocks.2.* | 5000 | | | | 1e-3 | 1e-3 |
| Phase 5 | Phase 4 + classifier.blocks.3.* | 5000 | | | | 1e-3 | 1e-3 |
| Phase 6 | Phase 5 + classifier.blocks.4.* | 5000 | | | | 1e-3 | 1e-3 |
| Phase 7 | Phase 6 + classifier.blocks.5.* | 5000 | | | | 1e-3 | 1e-3 |
| Phase 8 | all layers | 5000 | | | | 1e-3 | 1e-3 |
| **CCT7_3x1_32 @CIFAR10 (FDF strategy)** | | | | | | step_a | step_c |
| Phase 1 | tokenizer.conv_layers+classifier. positional_emb+attention_pool | 500 | | | | 2e-3 | 2e-3 |
| Phase 2 | Phase 1 + classifier.blocks.0.* | 500 | | | | 2e-3 | 2e-3 |
| Phase 3 | Phase 2 + classifier.blocks.1.* | 500 | train until loss<0.05 or $r$>1000 | 128 | SGD($\eta$=0.001, $\beta$=0) | 2e-3 | 2e-3 |
| Phase 4 | Phase 3 + classifier.blocks.2.* | 500 | | | | 2e-3 | 2e-3 |
| Phase 5 | Phase 4 + classifier.blocks.3.* | 500 | | | | 2e-3 | 2e-3 |
| Phase 6 | Phase 5 + classifier.blocks.4.* | 500 | | | | 2e-3 | 2e-3 |
| Phase 7 | Phase 6 + classifier.blocks.5.* | 500 | | | | 2e-3 | 2e-3 |
| Phase 8 | Phase 6 + classifier.blocks.6.* | 500 | | | | 2e-3 | 2e-3 |
| Phase 9 | all layers | 1000 | | | | 2e-3 | 2e-3 |
| **DLA @CIFAR10 (FDF strategy)** | | | | | | step_a | step_c |
| Phase 1 | base.0.* | 400 | | | | 1e-3 | 2e-3 |
| Phase 2 | Phase 1 + base.1.* | 400 | | | | 1e-3 | 2e-3 |
| Phase 3 | Phase 2 + layer1.* | 400 | | | | 1e-3 | 2e-3 |
| Phase 4 | Phase 3 + layer2.* | 400 | train until loss<0.005 or $r$>1000 | 128 | SGD($\eta$=0.001, $\beta$=0) | 1e-3 | 2e-3 |
| Phase 5 | Phase 4 + layer3.left_node.* | 400 | | | | 1e-3 | 2e-3 |
| Phase 6 | Phase 5 + layer3.right_node.* | 400 | | | | 1e-3 | 2e-3 |
| Phase 7 | Phase 6 + layer3.root.* | 400 | | | | 1e-3 | 2e-3 |
| Phase 8 | Phase 7 + layer4.prev_root.* | 400 | | | | 1e-3 | 2e-3 |
| Phase 9 | Phase 8 + layer4.level_1.* | 400 | | | | 1e-3 | 2e-3 |
| Phase 10 | Phase 9 + layer4.left_node.* | 400 | | | | 1e-3 | 2e-3 |
| Phase 11 | Phase 10 + layer4.right_node.* | 400 | | | | 1e-3 | 2e-3 |
| Phase 12 | Phase 11 + layer4.root.* | 400 | | | | 1e-3 | 2e-3 |
| Phase 13 | Phase 12 + layer5.prev_root.* | 400 | | | | 1e-3 | 2e-3 |
| Phase 14 | Phase 13 + layer5.level_1.* | 400 | | | | 1e-3 | 2e-3 |
| Phase 15 | Phase 14 + layer5.left_node.* | 400 | | | | 1e-3 | 2e-3 |
| Phase 16 | Phase 15 + layer5.right_node.* | 400 | | | | 1e-3 | 2e-3 |
| Phase 17 | Phase 16 + layer5.root.* | 400 | | | | 1e-3 | 2e-3 |
| Phase 18 | Phase 17 + layer6.left_node.* | 400 | | | | 1e-3 | 2e-3 |
| Phase 19 | Phase 18 + layer6.right_node.* | 400 | | | | 1e-3 | 2e-3 |
| Phase 20 | Phase 19 + layer6.root.* | 400 | | | | 1e-3 | 2e-3 |
| Phase 21 | all layers | 400 | | | | 1e-3 | 2e-3 |

[a] short for Algorithm 1 Line 4.

[b] the loss value is computed using a rolling average with a window size of 10.

[c] all layers in Phase 2.

[d] all layers in the first bottleneck layer.

Table 3: Hyperparameters used for applying Algorithm 2 to find the low-loss path toward origin.

| | Applied layers | Iteration T (A1 L4[7]) | Train round $r$ (A1 L7) | Batch size $\mathcal{B}$ (A1 L7) | Optimizer and hyperparameter x (A1 L7) | step_a (A1 L14) | step_c (A1 L14) |
|---|---|---|---|---|---|---|---|
| LeNet5 @MNIST | all layers | 1400 | train until loss <0.02 or $r$ >100 | 64 | SGD($\eta$=0.001, $\beta$=0) | 1e-3 | 0 |
| ResNet18 @CIFAR10 | all except norm layers | 1400 | train until loss <0.02 or $r$ >5000 | 256 | SGD($\eta$=0.001, $\beta$=0) | 1e-3 | 0 |
| ResNet18 @CIFAR100 | all except norm layers | 1400 | train until loss <0.02 or $r$ >5000 | 256 | SGD($\eta$=0.001, $\beta$=0) | 1e-3 | 0 |
| CCT7_3x1_32 @CIFAR10 | all except norm layers | 1000 | train until loss <0.05 or $r$ >5000 | 128 | SGD($\eta$=0.001, $\beta$=0) | 1e-3 | 0 |
| DLA @CIFAR10 | all except norm layers | 1000 | train until loss <0.005 or $r$ >1000 | 128 | SGD($\eta$=0.001, $\beta$=0) | 5e-4 | 1e-3 |

Table 4: Hyperparameters used to get a low-loss mode.

| | Training batch size | Optimizer and hyperparameters | Epoch | Learning rate $\eta$ (scheduler) |
|---|---|---|---|---|
| LeNet5 @MNIST | 64 | SGD($\beta$=0.9) | 20 | 0.01 |
| ResNet18 @CIFAR10 | 256 | SGD($\beta$=0.9, $\lambda$=5e-4) | 30 | OneCycle LR (Smith & Topin, 2018) max lr=0.1 |
| ResNet18 @CIFAR10 (used in Figure 6) | 256 | SGD($\beta$=0.9, $\lambda$=5e-4) | 140 | OneCycle LR max lr=0.1 |
| ResNet18 @CIFAR100 | 256 | SGD($\beta$=0.9, $\lambda$=5e-4) | 50 | OneCycle LR, max lr=0.1 |
| CCT7_3x1_32 @CIFAR10 | 128 | AdamW($\beta_1$=0.9, $\beta_2$=0.999,$\lambda$=6e-2) | 300 | Cosine Annealing LR (Loshchilov & Hutter, 2017) initial_lr=55e-5, warmup_lr=1e-5 min_lr=1e-5, warmup_epochs=10 cooldown_epochs=10 |
| MobileNet-V2 @CIFAR10 | 128 | SGD($\beta$=0.9, $\lambda$=4e-5) | 200 | MultiStep LR, milestone=100, gamma=0.1 max lr=0.1 |
| Regnet_x_200_mf @CIFAR10 | 256 | SGD($\beta$=0.9, $\lambda$=1e-4) | 120 | Cosine Annealing LR, initial_lr=0.1, warmup_epochs=0, min_lr=0, cooldown_epochs=0 |
| ShuffleNetv2 @CIFAR10 | 256 | SGD($\beta$=0.9, $\lambda$=4e-5) | 300 | MultiStep LR, milestone=[150,225], gamma=0.1 max lr=0.1 |
| VGG @CIFAR10 | 256 | SGD($\beta$=0.9, $\lambda$=1e-4) | 120 | Cosine Annealing LR, initial_lr=0.1, warmup_epochs=0, min_lr=0, cooldown_epochs=0 |
| DenseNet @CIFAR10 | 256 | SGD($\beta$=0.9, $\lambda$=1e-4) | 120 | Cosine Annealing LR, initial_lr=0.1, warmup_epochs=0, min_lr=0, cooldown_epochs=0 |
| EfficientNet-B0 @CIFAR10 | 256 | SGD($\beta$=0.9, $\lambda$=1e-4) | 120 | Cosine Annealing LR, initial_lr=0.1, warmup_epochs=0, min_lr=0, cooldown_epochs=0 |
| DLA @CIFAR10 | 256 | SGD($\beta$=0.9, $\lambda$=1e-4) | 120 | Cosine Annealing LR, initial_lr=0.1, warmup_epochs=0, min_lr=0, cooldown_epochs=0 |
| DLA @CIFAR100 | 256 | SGD($\beta$=0.9, $\lambda$=1e-4) | 120 | Cosine Annealing LR, initial_lr=0.1, warmup_epochs=0, min_lr=0, cooldown_epochs=0 |
| Hyperparameters used to find modes on difference variance sphere, see Figure 4 and Figure 13. One mode is obtained using the hyperparameters listed above, and the other is obtained using the hyperparameters listed below. | | | | |
| DLA @CIFAR10 | 256 | SGD($\beta$=0.9, $\lambda$=5e-4) | 120 | Cosine Annealing LR, initial_lr=0.2, warmup_epochs=0, min_lr=0, cooldown_epochs=0 |
| ResNet18 @CIFAR10 | 256 | SGD($\beta$=0.9, $\lambda$=1e-4) | 30 | OneCycle LR max lr=0.2 |
| CCT7_3x1_32 @CIFAR10 | 128 | AdamW($\beta_1$=0.9, $\beta_2$=0.999,$\lambda$=1e-2) | 300 | Cosine Annealing LR initial_lr=100e-5, warmup_lr=1e-5 min_lr=1e-5, warmup_epochs=10 cooldown_epochs=10 |
| Hyperparameters used to find sharp modes, used in Appendix A.9. | | | | |
| ResNet18 @CIFAR10 | 256 | SGD($\beta$=0.9, $\lambda$=0) | 30 | lr=0.1 |

Table 5: Dataset preprocessing and augmentation steps. The name of each step aligns with the corresponding function names used in PyTorch. The resulting samples are used for training low-loss models and in the *Train* step in Algorithm 1 and Algorithm 2.

| Data pre-precessing (augmentation) steps | Step 1 | Step 2 | Step 3 |
|---|---|---|---|
| MNIST | Random-rotate for 5 degrees | Random-crop to 28x28 with padding size 2 | Normalize with dataset mean (0.1307) and standard derivation (0.3081) |
| CIFAR10 | Random-crop to 32x32 with padding size 4 | Random-horizontal-flip with probability 0.5 | Normalize with mean [0.491,0.482,0.446] and standard derivation [0.247,0.243,0.262] |
| CIFAR100 | Random-crop to 32x32 with padding size 4 | Random-horizontal-flip with probability 0.5 | Normalize with mean [0.507,0.487,0.441] and standard derivation [0.268,0.257,0.276] |

Table 6: Computation time required to execute Algorithm 1 and Algorithm 2 on our system.

| Simulation | Algorithm 1 equivalent training dataset epoch (best case / worst case) | Algorithm 1 reference time | Algorithm 2 equivalent training dataset epoch (best case / worst case) | Algorithm 2 reference time |
|---|---|---|---|---|
| ResNet18 @CIFAR10 | 3584 / 35840 | 21 hours (Nvidia H100) | 35.84 / 358400 | 3 hours (Nvidia RTX 3090) |
| CCT7_3x1_32 @CIFAR10 | 1024 / 1024000 | 24 hours (Nvidia H100) | 2560 / 25600 | 25 hours (Nvidia RTX 3090) |
| DLA@CIFAR10 | 4608 / 46080 | 45 hours (Nvidia RTX 5090) | 512 / 5120 | 1 hour (Nvidia RTX 5090) |

As a comparison, we also evaluated AutoNEB in terms of both path quality and runtime on ResNet20@CIFAR10; see Figure 7. The AutoNEB authors provide two configurations—*standard* and *fast*—and we report results for both. Since AutoNEB does not guarantee successful mode connection, the maximum (saddle) loss along the path varies substantially across runs. Overall, LLPF attains significantly lower maximum training loss (approximately 0.05 in Figure 2, ResNet18@CIFAR10 panel) than AutoNEB. The configurations used in our comparison are listed in the footnote of Table 7.

The runtimes of FGE and SPRO are omitted, as these methods are designed for model ensembling and return only a single point on the connection rather than constructing an entire path.

Table 7: Computation time required to connect two modes using AutoNEB and the resulting mode-connection quality. The runtime and path quality of LLPF are reported in the final row for comparison.

| Simulation | AutoNEB runtime | saddle loss (maximum training loss) along the path |
|---|---|---|
| ResNet20@CIFAR10 | 6.5 hours (Nvidia H100) | 0.519 |
| ResNet20@CIFAR10 | 6.5 hours (Nvidia H100) | 1.648 |
| ResNet20@CIFAR10 | 6.4 hours (Nvidia H100) | 0.542 |
| ResNet20@CIFAR10 | 6.2 hours (Nvidia H100) | 1.541 |
| ResNet20@CIFAR10(fast) | 1.9 hours (Nvidia H100) | 4.200 |
| ResNet20@CIFAR10(fast) | 1.9 hours (Nvidia H100) | 1.240 |
| ResNet20@CIFAR10(fast) | 1.9 hours (Nvidia H100) | 2.898 |
| ResNet20@CIFAR10(fast) | 1.9 hours (Nvidia H100) | 2.630 |
| **ResNet18@CIFAR10(our)** | **21 hours (Nvidia H100)** | **0.04** |

*Notes:* configuration files are provided by AutoNEB authors and are available in `https://github.com/fdraxler/PyTorch-AutoNEB/blob/master/configs/cifar10-resnet20.yaml` and `https://github.com/fdraxler/PyTorch-AutoNEB/blob/master/configs/cifar10-resnet20-fast.yaml`.

### A.6 ADDITIONAL RESULTS FOR CONNECTING MODES ON DIFFERENT VARIANCE SPHERE

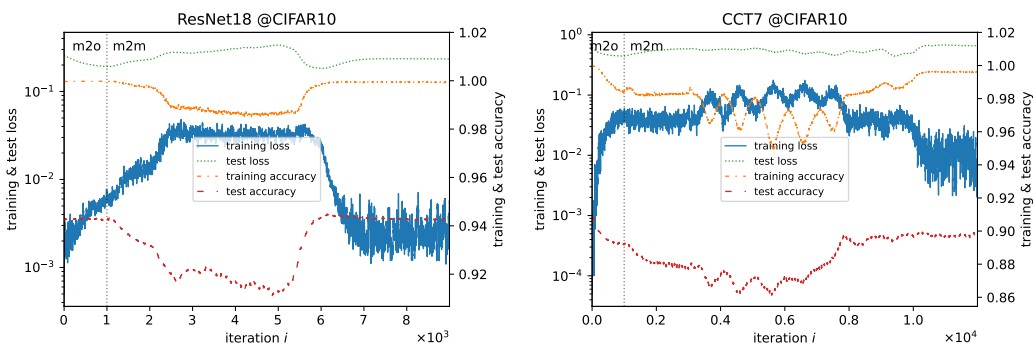

Figure 13: Extensions of the results in Figure 4 for ResNet18@CIFAR10 and DLA@CIFAR10 cases.

### A.7 ADDITIONAL RESULTS FOR THE TRAINING LOSS OF LINEAR INTERPOLATION

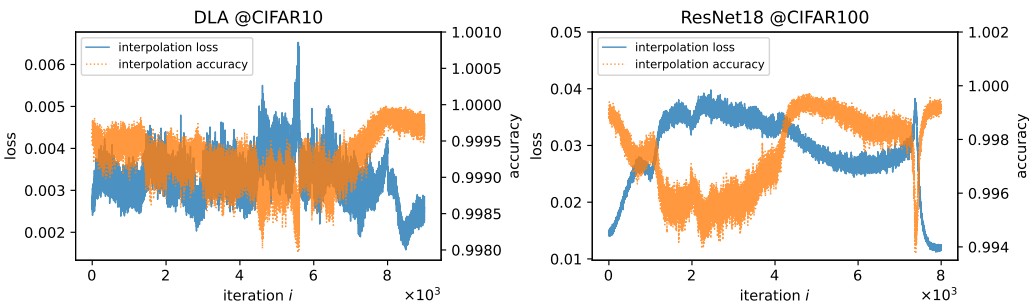

Figure 14: Extensions of the results in Figure 5 for CCT7@CIFAR10 and ResNet18@CIFAR100 cases.

### A.8 EMPIRICAL INVESTIGATION OF THE VARIANCE SPHERE RANGE SUPPORTED BY ALGORITHM 2

The destination point of Algorithm 2 is the origin, which is not a low-loss mode. Consequently, the training loss is expected to increase along the path identified by Algorithm 2, and beyond a certain point the path should no longer be considered low-loss. This raises a natural concern: if the training loss increases too quickly, the algorithm may exceed the loss threshold before reaching the target variance sphere, leading to the question whether Algorithm 2 can reliably connect modes located on different variance spheres.

In this section, we show that the region in which Algorithm 2 can identify low-loss paths is substantially larger than the region in which standard SGD is able to find low-loss solutions. To demonstrate this, we consider the ResNet18@CIFAR10 setting. We first train a reference mode, denoted $S$, using zero weight decay, and then train a sequence of additional modes with gradually increasing weight decay from $1 \times 10^{-5}$, $2 \times 10^{-5}$ to $4.096 \times 10^{-2}$. We apply Algorithm 2 to construct a low-loss path from $S$ to the origin, and we record the relationship between the variance of the first layer and the training loss along this path. We compare this relationship with that of the SGD solutions, see Figure 15.

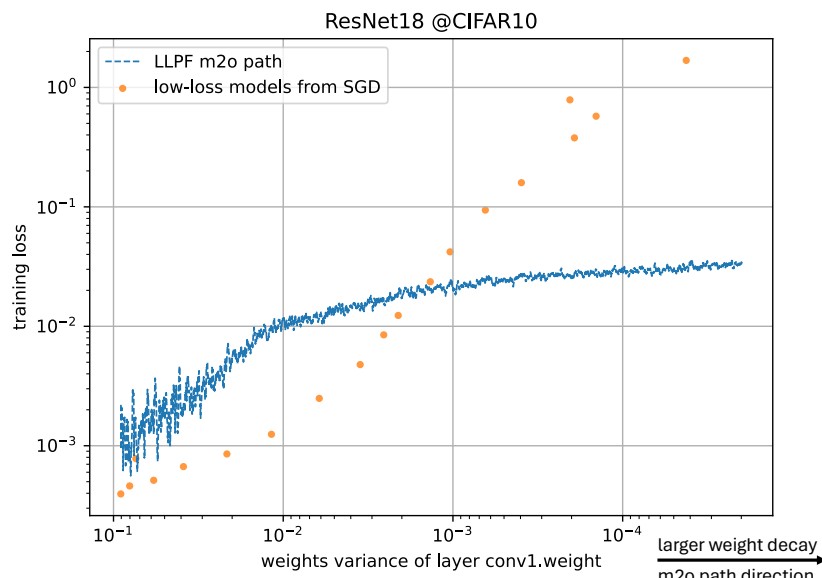

Figure 15: Relationship between the weight variance of the first layer (ResNet18) and the training loss. The x-axis uses the variance of the first layer as a proxy for the overall model weight variance, and the y-axis shows the training loss for both SGD solutions and points along the m2o path. The blue curve shows the m2o path found by Algorithm 2, while the orange points represent low-loss models obtained by SGD. The training loss along the m2o path increases much more slowly once it exceeds $0.01$, because the training round $r$ is increased when the loss approaches $0.02$, see Table 3.

The hyperparameters used to find this m2o path are identical to those in Table 3, except that the number of iterations $T$ is set to 4000 in order to obtain a much longer path. Figure 15 shows that the m2o algorithm can explore a substantially wider region of low variance while still maintaining low loss, whereas SGD cannot, due to the large weight decay. This result provides empirical evidence that m2o is effective for connecting modes obtained by SGD in general.

A.9 CONNECTING MODES LOCATED IN SHARPER MINIMA

In this section, we use the ResNet18@CIFAR10 setting to illustrate that Algorithm 1 can also connect modes located in sharp minima. To obtain a sharp mode, we train the model using a constant learning rate of 0.1 and zero weight decay. Compared with standard training hyperparameters, this configuration removes both model regularization techniques and learning-rate scheduling, and therefore the resulting modes are expected to lie in sharper minima. The detailed training hyperparameters are listed in Table 4.

To verify that the sharp mode is indeed sharper than the normal mode, we employ two procedures to quantify and visualize loss landscape sharpness. The first metric is relative flatness (Petzka et al., 2021), which measures how much the loss increases when model parameters are perturbed along directions aligned with the model's learned feature geometry. The second metric visualizes the loss landscape by randomly perturbing model weights and observing the induced change in loss.

The second metric can be formally expressed as follows. Let $\theta \in \mathbb{R}^D$ be the vector of all trainable parameters, and the set of perturbation ratio be $\mathcal{R} = \{r_1, \ldots, r_M\}$ and the number of sampled directions be $K$. For each $r \in \mathcal{R}$ and each $k \in \{1, \ldots, K\}$:

$$u_{r,k} \sim \text{random unit vector in } \mathbb{R}^D,$$
$$\theta'_{r,k} = \theta + r \, \|\theta\|_2 \, u_{r,k},$$
$$\Delta\mathcal{L}(r,k) = \left| \mathcal{L}(\theta'_{r,k}) - \mathcal{L}(\theta) \right|.$$

The sharpness profile at ratio $r$ is then

$$S(r) = \frac{1}{K} \sum_{k=1}^{K} \Delta\mathcal{L}(r, k),$$

Figure 16 compares the sharpness profiles of a normal mode and a sharp mode measured with $K = 100$ and $r = [1, 2, 3 \ldots 50] \times 10^{-2}$. The relative flatness results, computed according to Definition 3 in Petzka et al. (2021), are reported in the panel titles. We then generate a second sharp mode and apply Algorithm 1 using the hyperparameters in Table 2 to connect the two sharp minima. The resulting low-loss path is shown in Figure 17.

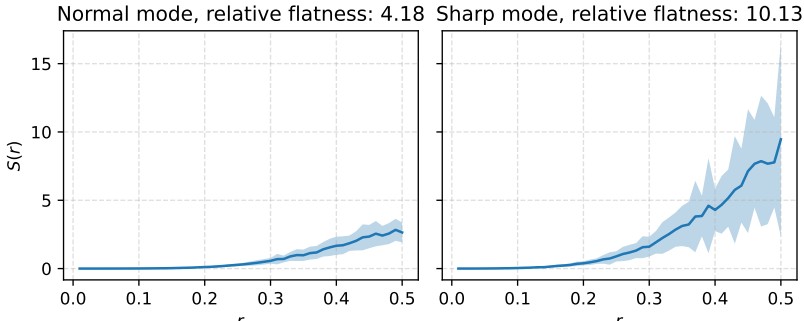

Figure 16: Sharpness of a normal mode (left) and a sharp mode (right). The curves show the mean sharpness values across $K = 100$ repetitions, and the shaded region denote the standard deviation. The relative flatness values are reported in the panel titles, where larger values correspond to sharper loss landscapes.

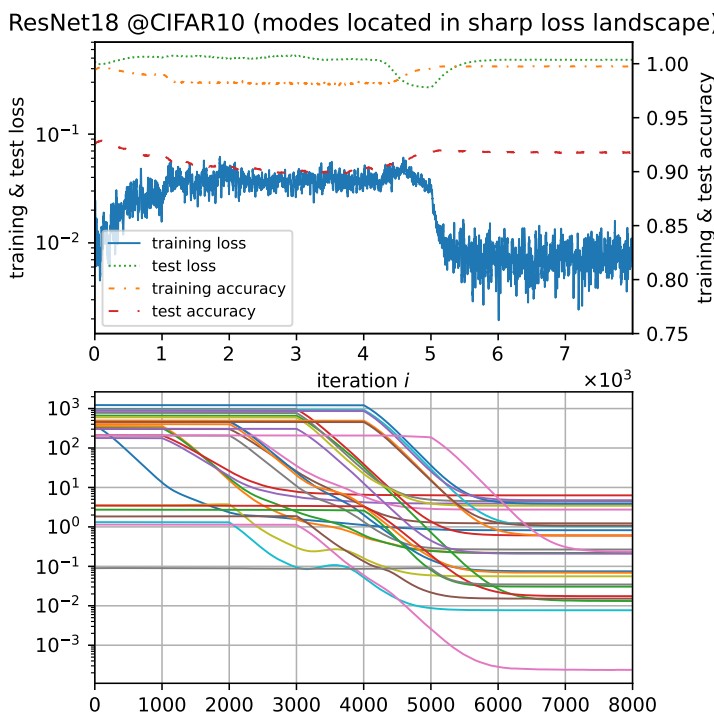

Figure 17: Connection of two modes located in sharp minima. The hyperparameters are selected according to the FDF strategy. The upper panel shows the accuracy and loss on the training and testing datasets, and the lower panel shows the layer-wise weight differences.

## A.10 PROOF OF EQUATION 6

**Proof of $\|\overrightarrow{OP_{lx}}\|^2 \propto \mathrm{Var}(\theta_{lx})$ (approximately):**

$$\mathrm{Var}(x) = \frac{1}{n} \sum_{i=1}^{n} (x_i - \bar{x})^2$$

where $n$ is the number of parameters in layer $l_x$, $x_i$ denotes the $i$-th parameter, and $\bar{x}$ is the mean of the parameters. Assuming Equation 5 holds, i.e., $\bar{x} \approx 0$, we have:

$$\mathrm{Var}(x) \approx \frac{1}{n} \sum_{i=1}^{n} x_i^2$$

$$n \cdot \mathrm{Var}(x) \approx \sum_{i=1}^{n} x_i^2$$

By definition, the squared Euclidean distance from the origin to point $P_{lx}$ is:

$$\|\overrightarrow{OP_{lx}}\|^2 = \sum_{i=1}^{n} x_i^2$$

Combining the two expressions, we obtain:

$$\|\overrightarrow{OP_{lx}}\|^2 \approx n \cdot \mathrm{Var}(x)$$

Since $n$ is a constant for a given layer, it follows that:

$$\|\overrightarrow{OP_{lx}}\|^2 \propto \mathrm{Var}(\theta_{lx})$$

which completes the proof.

## A.11 MODEL SELECTIONS AND IMPLEMENTATIONS

We select model architectures based on their popularity within the PyTorch community. Specifically, we consider all architectures with official pretrained weights on the PyTorch website that were introduced after ResNet and are suitable for the CIFAR dataset. Models that only support ImageNet-scale inputs (Deng et al., 2009) are excluded due to computational resource constraints. Following this criterion yields MobileNet, ShuffleNet, EfficientNet, and RegNet. For transformer-based models, vanilla ViT variants (Dosovitskiy et al., 2021) typically require ImageNet-level resolution and are therefore unsuitable for CIFAR; accordingly, we select CCT (Compact Convolutional Transformers) as a CIFAR-compatible transformer alternative.

To adapt ResNet18 to the image resolution of the CIFAR dataset, the first convolutional layer is modified to use a $3 \times 3$ kernel with stride 1 and padding 1. The implementations of ShuffleNet and MobileNet-V2 for the CIFAR10 dataset are obtained from (Chenhang, b;a). The implementations of VGG-11, DenseNet, EfficientNet-B0, Regnet_x_200_mf are obtained from (Kuangliu). For all other cases, the architecture remains identical to the original implementation.

## A.12 LLM USAGE DISCLOSURE

This manuscript was polished with the assistance of a large language model (LLM). The initial draft was written by the authors without LLM usage, and all results were obtained from experiments without LLM usage. The LLM was also used to generate plotting scripts based on the raw experimental results.

