# OpenReview forum: "Connecting Independently Trained Modes via Layer-Wise Connectivity"
_ICLR.cc/2026/Conference — Submitted to ICLR 2026_

### Official Review · Reviewer_suKz · 2025-10-21

**Soundness:** 3
**Presentation:** 2
**Contribution:** 2
**Rating:** 4
**Confidence:** 3

**Summary:**

This paper proposes an algorithm for finding low-loss paths between independently trained models across a wide variety of architectures. The proposed method is based on the empirical observation that the variance of trained model parameters (i.e., the sum of squared parameter values) tends to remain approximately constant. Using this approach, the authors demonstrate that continuous paths with small training losses can be found in various architectures trained on CIFAR-10. They also show that low-loss paths can be identified even between models trained with different hyperparameters, which result in different parameter variances.

**Strengths:**

* The paper proposes an algorithm that enables verification of mode connectivity across a variety of architectures.
* It demonstrates that low-loss paths can be discovered even between models trained with different hyperparameters.

**Weaknesses:**

* It is unclear whether Algorithm 2 can always successfully discover a path between models with different parameter variances.
* The experiments rely almost entirely on CIFAR-10.
* The method does not necessarily yield paths with low test loss.
* Although the paper claims that paths can be found between models trained with different hyperparameters, the actual experiments are restricted to a very limited setting.

**Questions:**

## The mathematical notation could be clarified to avoid confusion.

For example, $l_x$ is used to denote the $x$-th layer. However, $x$ is also used to represent a hyperparameter, and this dual use may cause confusion. In addition, $x$ commonly represents the input, so using it as a hyperparameter symbol is unconventional.

Furthermore, the variance of the $l_x$-th layer is written as $\mathrm{Var}(\theta_{l_x})$. However, this expression does not refer to the variance of a distribution in the usual statistical sense but rather to the squared L2-norm of the layer parameters. Using the term “variance” might imply that $\theta_{l_x}$ is a random variable drawn from a distribution determined by the training algorithm. Since the intent seems to be to compute the variance across the dimensions of $\theta_{l_x}$ itself, it would be clearer to simply write $\Vert \theta_{l_x} \Vert^2$.

## It is unclear whether Algorithm 2 consistently works as intended.

As I understand it, Algorithm 2 searches for a path from a model with larger parameter variance to another with smaller variance. It is used to align the variances of two pretrained models so that Algorithm 1 can then find a low-loss path between them. In other words, if Algorithm 2 fails to produce an intermediate model whose variance matches that of the destination model, Algorithm 1 may not work properly.

At line 402, the authors write:
> Unlike the results in Figure 2, the L_2 distance here is not expected to converge to very small values, because the destination point in Algorithm 2 is the origin, which is not itself a low-loss mode.

This suggests that Algorithm 2 does not always produce models with small L2 norms. This might be acceptable if we could guarantee that models with small parameter norms necessarily yield large losses. However, that is not always the case. For example, when a convolutional layer is followed by layer normalization, scaling the convolution weights does not change the layer output. Consequently, the variance of such weights could be arbitrarily small without degrading performance. Therefore, if one of the pretrained models happens to have a much smaller parameter norm, the proposed method might fail. Unless Algorithm 2 can **always** find the lowest-variance model with performance comparable to the original, this remains a potential limitation.

## The experiments are biased toward CIFAR-10.

Although various architectures are tested, the dataset choice is limited to CIFAR-10. It would be more convincing to include experiments on larger-scale datasets such as CIFAR-100 or ImageNet.

## There is no guarantee that models with low test loss will be found.

As acknowledged by the authors, the proposed algorithm ensures low training loss but not low test loss. If the purpose of this study is to provide new insight into the generalization properties of SGD, this limitation should be considered a notable weakness.

## It would be helpful to include experiments where hyperparameters other than weight decay are changed.

The paper claims that low-loss paths can be found between models trained with different hyperparameters. However, in practice, the only hyperparameter varied is the weight decay, which mainly affects the scale of the parameters. It is well known that the sharpness of the loss landscape is also influenced by other hyperparameters such as the learning rate. At line 314, the authors mention:
> Empirically, we find that starting from sharp minima can cause our algorithm to fail to find feasible paths.

However, the paper does not provide experiments that explicitly examine the effect of sharpness. It would strengthen the work to include experiments where other hyperparameters, such as the learning rate, are varied to demonstrate the broader applicability of the proposed method.

---

> ### Author Response · Authors · 2025-11-20
>
> Thank you for the time spent reviewing our paper. We address all your questions below, and we hope our new results can address your concerns.
>
> - The mathematical notation could be clarified to avoid confusion.
>
> Thank you for pointing this out. We have fixed the symbol of training hyperparameter issue in Change list (9).
>
> Regarding the variance of layer weights, we would like to further clarify our choice. During weight initialization, parameter values are sampled from a distribution characterized by a mean and variance [1]. Thus, variance has a clear statistical meaning at initialization. We agree that this statistical interpretation becomes less direct after training. However, our algorithm relies on variance during training for a different reason: the projection back to the variance sphere is designed to maintain the trainability of intermediate models [2], and this property is linked to the variance of weights, not the weight norm.
>
> At the same time, to present the geometric meaning of the algorithm, we also need to consider the norm of the parameters. Because both perspectives are important, we use both variance and norms, and we show in the Appendix that they remain roughly correlated in practice.
>
> For these reasons, we would not recommend replacing variance with a notation such as $\|\| \theta \|\|$, since this would remove the meaningful distinction that variance provides in our algorithmic design.
>
>
>
> - It is unclear whether Algorithm 2 consistently works as intended.
>
> Thanks for this question, we realize that an extra appendix section is required to explain it, see Change list (11). Briefly, we evaluate the variance range of an m2o path generated by Algorithm 2 and compare it against the variance of modes obtained via SGD under different weight-decay settings (which place them on different variance spheres). We find that the m2o path covers nearly all low-loss modes obtained by SGD in terms of layer-wise weight variance. This indicates that, for models produced by SGD, Algorithm 2 reliably finds a continuous path between them.
>
> - The experiments are biased toward CIFAR-10.
>
> We agree with the reviewer that our current experiments are weighted toward CIFAR-10, with only two CIFAR-100 results (ResNet and DLA) included in Appendix A.4. To address this concern, we included additional experiments on MobileNet, ShuffleNet, EfficientNet, and RegNet using CIFAR-100 in the revision. The new results confirm Algorithm 2 still works for CIFAR-100. See Change list (8).
>
>
> - There is no guarantee that models with low test loss will be found. As acknowledged by the authors, the proposed algorithm ensures low training loss but not low test loss. If the purpose of this study is to provide new insight into the generalization properties of SGD, this limitation should be considered a notable weakness.
>
> Thanks for pointing this out. Our study does not aim to analyze or characterize the generalization properties of SGD-trained models. Instead, the only focus is constructing low-loss paths between independently-trained modes. Model generalization could be the next step following our research work.
>
> - It would be helpful to include experiments where hyperparameters other than weight decay are changed.
>
> We agree that exploring variations beyond weight decay would strengthen the experimental coverage. We have re-run experiments of "Connecting Modes on Different Variance Sphere" part to connect modes obtained with different learning rate and weight decay. The results show that our algorithms continue to work reliably in this setting. This update is reflected in Change List (10).
>
> Regarding the sharpness, we removed "Empirically, we find that starting from sharp minima can cause our algorithm to fail to find feasible paths." because we find that our algorithm could also work for sharp minima, see Change list (7).
>
>
>
> [1] Glorot, X., & Bengio, Y. (2010). Understanding the difficulty of training deep feedforward neural networks. In Proceedings of the Thirteenth International Conference on Artificial Intelligence and Statistics (pp. 249–256). PMLR.
>
>
> [2] The initial submission, second paragraph of Section "Low-Loss Path Finding Algorithm", "Next, the ***VarianceCorrection*** step projects $M_1$ back as $M_2$ onto the variance sphere of $P_0$ to counteract the vanishing variance problem, a phenomenon wherein averaging uncorrelated neural networks leads to reduced parameter variance, thereby hindering subsequent training efforts"

---

### Official Review · Reviewer_9MCz · 2025-10-27

**Soundness:** 2
**Presentation:** 3
**Contribution:** 1
**Rating:** 2
**Confidence:** 3

**Summary:**

This paper proposes LLPF, a new algorithm that implements mode connectivity, i.e. finding a low-loss path between two modes obtained by independent training.

**Strengths:**

- The presentation is clear and the paper is easy to follow.
- The experiment results verify the effectiveness of the proposed algorithm.

**Weaknesses:**

- I don't see why this problem is important. In my understanding, mode connectivity is more like a phenomenon that helps us better understand the loss landspace, instead of a challenge that needs to be solved by an algortihm. Perhpas the authors can describe more of the practical usefulness of their algorithm?
- The algorithm looks trivial, and is not explained at all. For example, why do you need to project back to the variance sphere at each step? How do you ensure there is no loss barrier between the point and the projected point (e.g. M1 and M2, in the context of Figure 1)? how do you guarantee M3 is not too far away from M2?

**Questions:**

See Weaknesses.

---

> ### Author Response · Authors · 2025-11-20
> **Rebuttal (part 1)**
>
> Thank you for the time spent reviewing our paper. We address your questions below, following the order in which they were raised.
>
> - I don't see why this problem is important. In my understanding, mode connectivity is more like a phenomenon that helps us better understand the loss landspace, instead of a challenge that needs to be solved by an algortihm. Perhpas the authors can describe more of the practical usefulness of their algorithm?
>
> We have revised the Introduction section to better present the motivation behind our paper, see Change List (3). Briefly, our algorithm enables navigation in the low-loss region to further investigate the geometry properties of low-loss regions, which in turn contributes to a deeper understanding of the loss landscape.
>
> We agree that mode connectivity is a phenomenon that offers insights into the loss landscape. However, as we clarify in the initial submission, Lines 42–50 of the Introduction, "the difficulty of constructing such connections appears to increase with architectural complexity......whether mode connectivity persists in more recent and sophisticated models remains an open question". Existing work primarily evaluates ResNet-18, DenseNet, and VGG; whether mode connectivity persists in recent architectures remains an open question. Our contribution is to push this boundary to substantially more modern architecture families such as RegNet, DLA, and CCT.
>
> Thus, the problem is important because without a reliable algorithmic tool, one cannot determine whether a given model–dataset pair exhibits mode connectivity. Our method directly addresses this gap and enables new empirical insights into more recent architectures.
>
> Regarding practical usefulness, our algorithm enables the navigation of low-loss regions, which in turn provides deeper understanding of the loss landscape, as also noted by Reviewer WDuq. These investigations are not possible without a practical method for constructing the paths, and our algorithm provides that foundation.

---

> ### Author Response · Authors · 2025-11-26
> **Rebuttal (part 2)**
>
> - “The algorithm looks trivial… why project back to the variance sphere? How do you ensure there is no loss barrier between the point and the projected point (e.g. M1 and M2, in the context of Figure 1)? how do you guarantee M3 is not too far away from M2?"
>
> Thank you for the question. First of all, we would like to clarify that our algorithm is not trivial. Over the past seven years, only a small number of works have addressed mode connectivity between independently trained models, and among them, AutoNEB and SPRO works are the only empirical methods capable of constructing a full path. However, these methods have been demonstrated only on older architectures (ResNet, DenseNet) and leave open the question of whether mode connectivity persists in more recent architectures. (This point is discussed in the third paragraph of the Introduction in the initial submission.)
>
> Second, most of the relevant explanations are already available in the original submission, and we summarize them here.
>
> The reason of projecting back to the varaince sphere is available in Line 241 (initial submission): "projects $M_1$ back as $M_2$ onto the variance sphere of $P_0$ to counteract the vanishing variance problem, a phenomenon wherein averaging uncorrelated neural networks leads to reduced parameter variance, thereby hindering subsequent training efforts(Tian et al., 2024)."
>
> Third, loss barriers do not arise in our method for the following reasons:
>
> (1) Our trajectory does not pass through the mid-point where barriers typically occur.
> In AutoNEB, barriers appear at the midpoint of the linear interpolation between two modes (see Fig. 2 in Draxler et al. [1]). Our algorithm avoids this region because it constructs the path layer-wise, not by moving all parameters simultaneously. Therefore, the known barrier location is not visited.
>
> (2) Each step of our procedure is intentionally small.
> 1. The step $P_0M_1$ is bounded by the small step-size hyperparameters ($step_a$, $step_c$ in Table 3).
> 2. The training step $M_2M_3$ uses a limited number of training rounds(<5000) and a small learning rate (0.001), see Table 3.
> 3. The projections $M_1M_2$ and $M_3P_1$ are always smaller than their preceding step ($P_0M_1$ and $M_2M_3$). This is because projection distance is always the smallest distance among all connections to the point on the projection surface.
>
> Because each segment has small Euclidean distance, the loss cannot increase sharply, following the distance-based bounds of Neyshabur et al. [2].
>
> (3) We explicitly check for barriers.
> As described in Line 429 (initial submission), the "Mode-Connection Continuity Check" section evaluates the loss along the linear interpolation between every adjacent pair $P_i$ and $P_{i+1}$. If any barrier existed, it would appear here. Empirically, no such barriers are observed.
>
> [1] Felix Draxler, Kambis Veschgini, Manfred Salmhofer, and Fred Hamprecht. Essentially no barriers in neural network energy landscape. In Jennifer Dy and Andreas Krause (eds.), Proceedings of the 35th International Conference on Machine Learning, volume 80 of Proceedings of Machine Learning Research, pp. 1309–1318. PMLR, 10–15 Jul 2018. URL https://proceedings.mlr.press/v80/draxler18a.html.
>
> [2] Behnam Neyshabur, Srinadh Bhojanapalli, and Nathan Srebro. A pac-bayesian approach to spectrally-normalized margin bounds for neural networks, 2018. URL https://arxiv.org/abs/1707.09564

---

### Official Review · Reviewer_WDuq · 2025-10-31

**Soundness:** 3
**Presentation:** 4
**Contribution:** 3
**Rating:** 6
**Confidence:** 3

**Summary:**

The paper presents a novel method to find connecting low-loss paths between models by a layer-wise search. For each layer, the method explores the sphere of layer-parameters with similar variance. The method is sensitive to the order in which layers are traversed, arguing that the order should follow the data-flow. The method is applicable to modern architectures where standard linear or Bezier interpolation approaches fail.

**Strengths:**

- The proposed method to find low-loss paths is novel and interesting.
- The proposed method is tested for recent model architectures.
- The notion of a _variance sphere_ and the layer-wise variance correction step are intuitive and sound.
- The empirical evaluation shows the method works for a wide variety of model architectures and results are consistent across seeds.
- Algorithms are clearly presented and linked to geometric reasoning.
- Consistent paths across seeds and architectures suggest a shared geometric structure in modern networks, potentially revealing deeper properties of the loss landscape.

**Weaknesses:**

- The experiments are _somewhat_ limited to visualizations of loss/accuracy trajectories; no quantitative comparison of path quality (e.g., path length, interpolation efficiency, energy landscape visualization).
- The effect of layer order, variance correction, and training steps is not systematically analyzed in an ablation study.
- The approach is iterative and layer-wise. The paper acknowledges high compute requirements but gives no runtime analysis.

**Questions:**

- You state that “minima with near-zero training loss are generally flat.” Could you quantify this or connect it to measurable flatness metrics (e.g., relative flatness [3], Fisher-Rao Norm [2])?
- The observation that low training loss implies flatness can be explained for the CE loss: Walter et al. [4] have shown that flatness relates to model confidence and model confidence increase wit ha decreasing CE loss.
- Have you checked whether sharp minima with near-zero loss actually fail your algorithm? That would support your empirical claim.
- Since the paper’s method depends on starting from flat minima, and relative flatness [3] formalizes layer-wise curvature-weight norms, there seems to be a conceptual overlap. Could the variance-sphere assumption be seen as implicitly fixing relative flatness per layer?
- Adilova et al. [1] found that certain layers, or layer clusters, have a stronger impact on the loss barrier. How would ordering the layers according to their _importance_, i.e., influence on the loss barrier, impact your method?
- Adilova et al. [1] also found that different directions in parameter space have different impacts on the loss value. It would be interesting to see whether the layer-wise updates of the proposed methods correspond to the space perpendicular to the training space and averaging direction.

References:

[1] Adilova, Linara, et al. "FAM: Relative Flatness Aware Minimization." Topological, Algebraic and Geometric Learning Workshops 2023. PMLR, 2023.

[2] Liang, Tengyuan, et al. "Fisher-rao metric, geometry, and complexity of neural networks." The 22nd international conference on artificial intelligence and statistics. PMLR, 2019.

[3] Petzka, Henning, et al. "Relative flatness and generalization." Advances in neural information processing systems 34 (2021): 18420-18432.

[4] Walter, Nils Philipp, et al. "When Flatness Does (Not) Guarantee Adversarial Robustness." arXiv preprint arXiv:2510.14231 (2025).

---

> ### Author Response · Authors · 2025-11-20
> **Rebuttal (part 1)**
>
> Thank you for the time spent reviewing our paper. We address your points below, ordered from weaknesses to questions.
>
> - The experiments are somewhat limited to visualizations of loss/accuracy trajectories; no quantitative comparison of path quality (e.g., path length, interpolation efficiency, energy landscape visualization).
>
> Thank you for the suggestion. While metrics such as path length or interpolation efficiency are interesting, they are outside the scope of our current work. Our primary goal is to demonstrate that the proposed algorithm reliably connects independently trained minima. For this purpose, training and test loss/accuracy along the path provide the most direct and sufficient evidence that the constructed path remains low-loss and contains no barriers.
>
> - The effect of layer order, variance correction, and training steps is not systematically analyzed in an ablation study.
>
> We agree that a systematic ablation analysis of each step would be valuable. However, we expect that such an analysis would require a substantial amount of additional experimentation and is beyond the scope of the current paper. At this stage, the central question for the area is whether we have a method that can reliably connect low-loss modes. Our work focuses on establishing this capability, and we view a comprehensive ablation study as an important future direction.
>
> - The approach is iterative and layer-wise. The paper acknowledges high compute requirements but gives no runtime analysis.
>
> During the reversion, we add the runtime of AutoNEB and compare the path quality, see Change List (2).
>
> We provide reference runtimes for our method in Appendix Table 6, along with the equivalent number of training epochs since training iterations account for most of the computational cost. We acknowledge that more detailed performance profiling could be conducted to analyze bottlenecks in the implementation, but we believe such profiling is outside the scope of the current paper.
>
> - You state that “minima with near-zero training loss are generally flat.” Could you quantify this or connect it to measurable flatness metrics (e.g., relative flatness [3], Fisher-Rao Norm [2])?
>
> Thank you for the suggestion. We performed additional experiments and found that our algorithm also works for sharp minima. To quantify sharpness in this setting, we use the relative flatness measure when connecting such minima. The corresponding results have been added to the revised version, see Change List (7).
>
>
> - The observation that low training loss implies flatness can be explained for the CE loss: Walter et al. [4] have shown that flatness relates to model confidence and model confidence increase with a decreasing CE loss.
>
> Thank you for the suggestion. We indeed find this work could support our empirical claim, and have added this citation into our revision, see Change List (6).
>
> - Have you checked whether sharp minima with near-zero loss actually fail your algorithm? That would support your empirical claim.
>
> We would like to especially thank the reviewer for this comment. Initially, we performed an experiment that was not presented in this paper. We trained two models using the Adam optimizer with poorly chosen hyperparameters and obtained a training loss of approximately 0.5. In this setting, Algorithm 1 consistently failed to identify feasible low-loss directions, which led us to believe that the method required starting from flat minima.
>
> Following the reviewer’s suggestion, we conducted more experiments with sharp minima that still achieve near-zero training loss. The details of this new experiment have been added in the revised version (see Change List (7)). Briefly speaking, we trained two models using a large constant learning rate and zero weight decay, which are known to reduce generalization and produce sharper solutions. We then applied Algorithm 1 to connect these two sharp minima, and the method succeeded.
>
>
>
> - Since the paper’s method depends on starting from flat minima, and relative flatness [3] formalizes layer-wise curvature-weight norms, there seems to be a conceptual overlap. Could the variance-sphere assumption be seen as implicitly fixing relative flatness per layer?
>
> Thanks for this question. Let me first clarify that our methods do not rely on starting from flat minima any more, we add a section in Appendix to present the experiment results of connect sharp minima, see Change List (7).
>
> From our understanding, projecting onto the variance sphere does not implicitly fix the relative flatness of each layer. This is because the dimension of variance-sphere is $R^{D-1}$, which is extremely large compared to the much lower-dimensional subspaces that would be required to preserve relative flatness, which depends on curvature–weight interactions rather than just global parameter variance.

---

> ### Author Response · Authors · 2025-11-20
> **Rebuttal (part 2)**
>
> - Adilova et al. [1] found that certain layers, or layer clusters, have a stronger impact on the loss barrier. How would ordering the layers according to their importance, i.e., influence on the loss barrier, impact your method?
>
> We appreciate this question and note that our empirical observations are consistent with those reported in Adilova et al. [1]. In our experiments, the order in which layers are moved plays a critical role. As mentioned in the paper, we recommend moving layers in the direction of data flow, from shallow to deeper layers. This recommendation is based on extensive experimentation, where deviations from this order frequently led to encountering loss barriers.
>
> Our empirical experience suggests that layers mutually influence one another, and the effect of a particular layer depends on which layers have already been updated. Because the influence of each layer changes dynamically during the process, we believe it is difficult to assign a static, per-layer “importance” score.
>
> We would also like to share an illustrative example. In the ResNet-18@CIFAR-10 experiment shown in Figure 2, all layers are updated simultaneously for the first 10000 ticks (hyperparameters available in Table 2). During this stage, the L2 distance decreases more and more slowly while the training loss remains high. This behavior indicates that the algorithm is encountering a loss barrier. Once we switch to moving layers according to data-flow order (hyperparameters after 10000 ticks), the training loss drops and the L2 distance begins decreasing again, even though only a single layer is being updated.
>
>
>
>
> - Adilova et al. [1] also found that different directions in parameter space have different impacts on the loss value. It would be interesting to see whether the layer-wise updates of the proposed methods correspond to the space perpendicular to the training space and averaging direction.
>
> We fully agree with the statement that different directions in parameter space have different impacts on the loss value. This is because the loss landscapes of neural network models are usually viewed as basins. Our empirical results suggest that there are in fact low-loss tunnels between these basins. This indicates that the choice of direction indeed matters.
>
> Regarding the question of whether layer-wise updates correspond to directions perpendicular to the training or averaging directions, we would like to clarify that perpendicularity is a trivial concept in high-dimensional parameter spaces. In very high dimensions, the probability that two randomly sampled directions are nearly orthogonal approaches one.
>
> Formally, consider two vectors $A=[a_1,a_2....a_D]$ and $B=[b_1,b_2....b_D]$. Their dot product is $A \cdot B = \sum_{i=1}^D a_ib_i$.
>
> When $D$ is large and the coordinates are drawn from distributions with zero mean and bounded variance, concentration phenomena imply that the dot product tends toward zero with high probability. As a result, almost all directions in the weight space become approximately perpendicular in high dimensions.
>
> We do believe that the layer-wise update mechanism interacts with the geometry of the low-loss region (training space), but this relationship is more complicated than perpendicularity.

---

### Official Review · Reviewer_WGfT · 2025-11-01

**Soundness:** 2
**Presentation:** 2
**Contribution:** 1
**Rating:** 2
**Confidence:** 3

**Summary:**

This paper proposes a new algorithm (Low Loss Path Finding) to mode connect two independently (from different initializations) trained (with different data ordering and augmentations) networks. The paper tests the algorithm on architectures that are not common in this line of work (MobileNet, ShuffleNet, etc), claiming that they are more recent. The authors claim that their method is more reliable—producing consistent paths for networks trained with different hyper-parameters---though they don’t actually cite or confirm if the prior work fail on these settings.

**Strengths:**

- The authors revisit non-linear mode connectivity to a broader set of architectures beyond the commonly studied ResNet/VGG models.
- I believe the discussion of the models trained with different hyper-parameters and the focus on different variances is an important point that is not addressed in the prior literature. The algorithm and the discussion of data flow order is sound.

**Weaknesses:**

- The paper does not provide clear motivation for why demonstrating mode connectivity in MobileNet, ShuffleNet, or the other selected architectures is important or beneficial. What practical or theoretical insights would we gain from showing mode connectivity in these specific models? The choice of architectures appears arbitrary and is not justified. Moreover, the proposed architectures (MobileNet, ShuffleNet, EfficientNet, RegNet) are roughly contemporary with the mode connectivity literature itself, contradicting the claim of addressing "modern" architectures. Addressing different model families like transformers, diffusion models could have been a fresher take on this topic.
- The architectures in the (linear) mode connectivity literature are indeed outdated. The paper should cite Juneja 2022, and Altıntaş 2025 for discussion of transformer-based models and NLP domain.
- Even though they claim to be more effective, the paper doesn’t compare the proposed method against the baseline methods (NEB, FGE, and SRPO) on the architectures that are evaluated in the prior work and the architectures that the paper expands in.
- The experimental setup appears to be inherited from prior work without significant extension or deepening. Even all three baselines (Garipov, Draxler, Benton) conducted experiments at the CIFAR-100 scale. The paper does not appear to push beyond the established experimental boundaries in terms of dataset scale, complexity, or diversity of evaluation metrics
- The paper claims the method "generalizes beyond traditional architectures" and supports a "broader range of networks," but the selected architectures are not particularly modern and the experimental validation does not convincingly demonstrate this generalization beyond what prior methods might achieve.
- The benefits of the algorithm is not validated. According to Table 6, algorithm 1 takes 21 hours on an H100 GPU for a ResNet18 on Cifar-10. If I am not misreading this table, the cost of the algorithm is quite high. Training a ResNet18 or 20 on Cifar-10 shouldn’t take more than 10 minutes.

**Questions:**

- Could the authors explain the different phases in L2 distance in Figure 2? Is there any pattern emerging? I am also curious if the authors have an insight about behavior in the top subfigure and its implications for the local geometry?
- Why were these specific architectures chosen? What are the practical implications of achieving mode connectivity for these specific architectures?
- Can the authors provide experimental results comparing their method to NEB, FGE, and SRPO on the same set of architectures? How does the proposed method's computational cost compare to these baselines? Table 6 could incorporate the runtime for these baseline methods.
- The authors could draw connection between the muP regime and their algorithm to provide insights. The paper could also benefit from discussing scale invariances.

---

> ### Author Response · Authors · 2025-11-20
> **Rebuttal (part 1)**
>
> Thank you for the time spent reviewing our paper. We address your points below, ordered from weaknesses to questions.
>
> - The paper does not provide clear motivation for why demonstrating mode connectivity in MobileNet, ShuffleNet, or the other selected architectures is important or beneficial. What practical or theoretical insights would we gain from showing mode connectivity in these specific models?
>
> Thank you for the comment. We have revised the motivation paragraph in the Introduction to clarify the practical and theoretical reasons for studying mode connectivity in these architectures. Please see Change List (3). Briefly, our algorithm enables investigation of the geometry and topology of low-loss regions, which in turn contributes to a deeper understanding of the loss landscape and neural network behavior.
>
> - The choice of architectures appears arbitrary and is not justified. Why were these specific architectures chosen?
>
> We would like to thank the Reviewer for reminding us to present the model selection procedure. We have added one paragraph in the Appendix "Model Selections and Implementations" to explain the selection.
> Our selection follows a clear and reproducible rule:
> (1) We select architectures based on popularity. More specifically, we consider all architectures with official pretrained weights on the PyTorch website (https://docs.pytorch.org/vision/main/models.html)
> (2) that were released after ResNet,
> (3) and that are suitable for CIFAR dataset.
>
> Following this criterion yields MobileNet, ShuffleNet, EfficientNet, and RegNet.
> For transformer-based models, vanilla ViT variants typically require ImageNet-scale resolution and are unsuitable for CIFAR[1]; therefore, we select CCT (Compact Convolutional Transformers) as a CIFAR-compatible transformer alternative.
>
> Running experiments at ImageNet scale is computationally prohibitive: a 24-hour CIFAR run would translate to several days per run on ImageNet, making hyperparameter tuning infeasible.
>
>
>
>
> - Moreover, the proposed architectures (MobileNet, ShuffleNet, EfficientNet, RegNet) are roughly contemporary with the mode connectivity literature itself, contradicting the claim of addressing "modern" architectures. Addressing different model families like transformers, diffusion models could have been a fresher take on this topic.
>
> We respectfully disagree. The reviewer under-estimates the difficulty of conencting independently-trained modes. As summarized in Table 1, existing mode-connectivity work still focuses on ResNet, VGG, DenseNet, and basic CNNs-all earlier than MobileNet, ShuffleNet, EfficientNet, RegNet, DLA and CCT. Therefore, the architectures we study are indeed more modern relative to the empirical scope of prior work.
>
> Furthermore, we do include a transformer-based model: CCT, and present it as one of the major results, see Fig. 2 and Fig. 3. The reason that we do not investigate diffusion models is identical to the reason we do not investigate Imagenet - not sufficient computational resources.
>
>
>
> - The architectures in the (linear) mode connectivity literature are indeed outdated. The paper should cite Juneja 2022, and Altıntaş 2025 for discussion of transformer-based models and NLP domain.
>
> We searched for the requested works and found the following:
> [Juneja 2022]: Jeevesh Juneja, Rachit Bansal, Kyunghyun Cho, João Sedoc, & Naomi Saphra. 2023. Linear Connectivity Reveals Generalization Strategies.
> [Altıntaş 2025]: Gul Sena Altintas, Gregor Bachmann, Lorenzo Noci, & Thomas Hofmann. 2023. Disentangling Linear Mode-Connectivity.
>
> The publication years do not match the reviewer’s citations exactly. If different papers were intended, we would appreciate clarification so that we can address them appropriately.
>
> [Juneja 2022]: focuses on transfer-learning scenarios (e.g., different basins under fine-tuning).
> As we explicitly state in Line 213, our work studies independently trained models, not transfer-learning area.
>
> [Altıntaş 2025]: also examines linear mode connectivity, again in settings distinct from ours; additionally, the paper is incomplete (4-page arXiv manuscript, experiments only on MNIST, MLP).
>
> Both works concern linear mode connectivity, whereas we study mode connections between independently trained modes, which is a distinct problem setting. As discussed in the section “Mode Connectivity in Spawning and Permutation,” linear mode connectivity represents a different line of research. For this reason, we believe direct comparison between these works and our work is not appropriate. That said, we will follow the reviewer’s suggestion and include Juneja et al. in our “Mode Connectivity in Spawning and Permutation” section to clarify the distinction between linear connectivity and connectivity among independently trained modes.
>
> We also note that our experiments already include a transformer-based model (CCT). Regarding NLP models, we agree this is an interesting future direction, but it is outside the scope of this paper.

---

> ### Author Response · Authors · 2025-11-20
> **Rebuttal (part 2)**
>
> - The experimental setup appears to be inherited from prior work without significant extension or deepening. Even all three baselines (Garipov, Draxler, Benton) conducted experiments at the CIFAR-100 scale. The paper does not appear to push beyond the established experimental boundaries in terms of dataset scale, complexity, or diversity of evaluation metrics
>
> We agree that our dataset scale does not exceed prior work, and we do not claim otherwise. The contributions (Line 63) of our proposed algorithm lie elsewhere: (1) supporting a broader range of architectures, (2) producing more consistent connectivity results, and (3) connecting models trained under different hyperparameters. Our experimental design is appropriate for evaluating these contributions.
>
>
>
> - The paper claims the method "generalizes beyond traditional architectures" and supports a "broader range of networks," but the selected architectures are not particularly modern and the experimental validation does not convincingly demonstrate this generalization beyond what prior methods might achieve.
>
> All architectures we evaluate (MobileNet, ShuffleNet, EfficientNet, DLA, RegNet, and CCT) are strictly more modern than those explored in prior mode-connectivity papers (ResNet, VGG, DenseNet, basic CNNs). Our algorithm successfully constructs low-loss paths for all these families, demonstrating stronger architectural generality.
>
> If the concern is whether prior methods might also work on these architectures, we agree this is theoretically possible. However, in past six years, no prior work has demonstrated such results, including Benton (2021), which still limits experiments to VGG and ResNet (Table 1).
> Our experiments provide the first such evidence.
>
>
>
> - The benefits of the algorithm is not validated. According to Table 6, algorithm 1 takes 21 hours on an H100 GPU for a ResNet18 on Cifar-10. If I am not misreading this table, the cost of the algorithm is quite high. Training a ResNet18 or 20 on Cifar-10 shouldn’t take more than 10 minutes.
>
> We are afraid this concern arises from a misunderstanding of our work. As stated in the title and in Line 34, our objective is to find a low-loss path between two independently trained models in $R^D$. The output of our method is therefore a path, not a single trained model.
>
> In the ResNet-18@CIFAR-10 example (Figure 2), the low-loss path consists of 35,000 intermediate low-loss models. Thus, comparing the runtime of path finding with the runtime of training a single model (10 minutes) is meaningless. Our method solves a different problem: constructing a continuous path connecting two modes, which inherently requires evaluating and updating many intermediate points.
>
> The benefits of our proposed algorithm is available in Table 1.
>
> - Could the authors explain the different phases in L2 distance in Figure 2? Is there any pattern emerging? I am also curious if the authors have an insight about behavior in the top subfigure and its implications for the local geometry?
>
> The patterns observed in the L2-distance panel of Figure 2 are directly induced by the hyperparameters used in Algorithm 1. As stated in Line 277, “The most critical hyperparameters for successfully finding a low-loss path using Algorithm 1 are the choice and order of layers, because the method operates one layer at a time, as the variance sphere is defined per layer.”
> Thus, the changes in L2 distance simply reflect which layers are being updated during each phase.
>
> Taking the ResNet-18 @ CIFAR-10 case as an example:
>
> In Phase 1 (Table 2), Algorithm 1 is applied to all layers from tick 0 to tick 10,000. Consequently, the L2 distances of all layers decrease simultaneously. The corresponding training-loss curve (top panel) shows a rapid increase during this stage, after which the loss stabilizes and the L2 distances plateau at an intermediate level.
>
> In Phase 2 (and subsequent phases), Algorithm 1 is applied only to specific subsets of layers, as listed in Table 2. The L2-distance panel exhibits distinct “segments” or “phases” that align with which layer groups are being updated.
>
> Regarding the top loss panel: the observed behavior is not due to geometric structure of the landscape, but follows from the same mechanism. Different layers have different functional roles, and updating different subsets of layers perturbs the loss by different magnitudes. The changes in the loss curve therefore correspond directly to which layers are being moved during each phase.
>
> - What are the practical implications of achieving mode connectivity for these specific architectures?
>
> Thanks for the suggestion, we have rewritten the paragraphs in Introduction to clarify our motivation, see Change list (3).  Briefly, our algorithm enables investigation of the geometry and topology of low-loss regions, which in turn contributes to a deeper understanding of the loss landscape and neural network behavior.

---

> ### Author Response · Authors · 2025-11-20
> **Rebuttal (part 3)**
>
> - Can the authors provide experimental results comparing their method to NEB, FGE, and SRPO on the same set of architectures?
>
> Thanks for the suggestion. In the revised work, we provide a comparison of the maximum training loss along the mode connections in Table 1 for ResNet architectures, where baselines such as NEB and FGE report results. Our method not only outperforms ResNet-20, which has comparable model complexity to our ResNet-18 setting, but also achieves a lower maximum training loss than SPRO, despite SPRO being evaluated on ResNet-158, a model with substantially more parameters.
>
> - How does the proposed method's computational cost compare to these baselines? Table 6 could incorporate the runtime for these baseline methods.
>
> Thank you for the suggestion. We have added the runtime of AutoNEB to Table 6 (see Change List (2)).
> Briefly, using the configuration file provided by the AutoNEB authors, AutoNEB reaches a maximum training loss of 0.50 in approximately 6.5 hours on an H100 GPU, whereas our algorithm achieves a maximum training loss of 0.04 in 21 hours. Our algorithm is slower but obtains a much better path.
>
>
>
> - The authors could draw connection between the muP regime and their algorithm to provide insights. The paper could also benefit from discussing scale invariances.
>
> Thanks for this suggestion. However, we do not see the relationship between muP regime and our paper. From our understanding, muP regime provides a framework for transferring training hyperparameters across model scales. Scale invariance, similarly, concerns architectural designs that preserve functional behavior under rescaling of weights or inputs, typically requiring specific layer types or activation functions. Our method does not involve transferring hyperparameters between models, nor does it rely on architectural design choices related to scale invariance. Instead, our focus is on constructing low-loss nonlinear paths between independently trained models, where the architectures and training setups remain fixed.
>
>
> [1] Ali Hassani, Steven Walton, Nikhil Shah, Abulikemu Abuduweili, Jiachen Li, & Humphrey Shi. (2022). Escaping the Big Data Paradigm with Compact Transformers.

---

### Author Response · Authors · 2025-11-20

Thank you for the time spent reviewing our paper. We appreciate the reviewers’ comments that would help us further improve the presentation of our work.

We address all reviewers' concerns in the revised version of our work. The major changes of our manuscript include:

- Motivation: We have rewritten Paragraphs 1, 2 and 3 in the Introduction to clarify the motivation behind our work.

- Dataset scale: we additionally conduct experiments for MobileNet, ShuffleNet, RegNet, and EfficientNet on CIFAR-100 in the revision. Our evaluation is conducted on DLA and ResNet with CIFAR-scale datasets in the original submission because connecting modes on ImageNet is computationally prohibitive.

- Architecture selection: The model architectures were selected based on their popularity within the PyTorch community. For transformer-based models, we use CCT as a practical alternative to ViT, which does not support CIFAR-scale inputs. We have added an explanation of this selection procedure in the Appendix.

- Results and Evaluation: we perform all experiments requested by the reviewers and include them in the revised version. For details please check the detailed change list.

- We conduct additional experiments to show that Algorithm 1 still works for sharp minima: In our earlier statement in the “Prerequisite” section claiming that “input models should lie in flat low-loss regions.” After conducting additional experiments, we found that Algorithm 1 also works for sharp minima. These new results are included in the Appendix under “Connecting modes located in sharper minima.” and we have revised the “Prerequisite” section to remove the flat-loss requirement. We would like to express our special thanks to Reviewer WDuq and Reviewer suKz for encouraging us to investigate and clarify this point.

We also have some general clarification of related works:

- Linear mode connectivity: we would like to address that ***mode connectivity across independently trained modes*** is different from ***linear mode connectivity (LMC)***. Please check our "Mode Connectivity in Spawning and Permutation" Section (Page 4).

- AutoNEB, FGE, and SPRO: We will compare our runtime against AutoNEB in Table 7. We do not include the runtimes of FGE (Fast Geometric Ensembling) or its follow-up method SPRO, because these approaches produce only an ensembled model, i.e., a single point on the mode connection, rather than constructing a full continuous path. We clarify this point in Table 1.


The changes of our manuscript are listed below.

Change list:

(1) Added citation [Juneja 2023] in the section “Mode Connectivity in Spawning and Permutation” as an example of linear mode connectivity in the transfer-learning setting. [Reviewer WGfT]

(2) Add the runtime of AutoNEB to Table 7. [Reviewer WGfT].

The runtime of FGE and SPRO is not included because these methods output only a single point, as they are model-ensembling approaches rather than full path-finding algorithms. As stated in the FGE paper [1], bottom of page 6: "While inspired by mode connectivity, FGE does not rely on explicitly finding a connecting curve...can be trained in the time required to train a single network." SPRO builds upon FGE to reason about mode-connecting volumes and accelerate ensembling, and therefore inherits this property.
We have added a sentence in Table 1 to make this clear to the reader.

(3) Revised the paragraph 1,2 and 3 in the Introduction to better explain the motivations behind our work. [Reviewer WGfT, Reviewer 9MCz]

(4) Added a clarifying sentence to the Figure 2 caption explaining the patterns observed in both the training-loss and L2-distance panels. [Reviewer WGfT]

(5) Added a paragraph in Appendix 11, detailing how we select model architectures. [Reviewer WGfT]

(6) Added a sentence in the “Prerequisite” section (Line 309) providing theoretical support for our claim that minima with near-zero loss are typically flatter. [Reviewer WDuq]

(7) Added a new Appendix section "Connecting modes located in sharper minima", evaluating whether Algorithm 1 succeeds on sharp minima with near-zero loss. Add ***relative flatness measure*** and classical sharpness measure. [Reviewer WDuq, Reviewer suKz]

(8) Conduct additional experiments on MobileNet, ShuffleNet, EfficientNet, and RegNet using CIFAR-100. [Reviewer suKz]

(9) The symbol for training hyperparameters are replaced with Greek letter 𝜉. [Reviewer suKz]

(10) Conduct experiments of "Connecting Modes on Different Variance Sphere" with different learning rate and weight decay. [Reviewer suKz]

(11) Add a section in Appendix "Empirical Investigation of the variance sphere range supported by Algorithm 2" to show that Algorithm 2 could cover most of the modes obtained by SGD. [Reviewer suKz]

[1] Timur Garipov et al, Loss Surfaces, Mode Connectivity, and Fast Ensembling of DNNs.

---

> ### Author Response · Authors · 2025-11-28
> **Additional changes**
>
> Change List:
>
> (12)  Since Reviewer WDuq noted that “our work can potentially reveal deeper properties of the loss landscape,” we fully agree that our empirical results point to such possibilities. To reflect this, we have added a Discussion section in the main text, after the Limitations section.
>
> Briefly, our empirical observations suggest two conjectures:
> For the model architecture and dataset pairs investigated in our paper:
> 1. Different isolated basins are connected by low-loss tunnels.
> 2. Because our algorithm consistently finds these tunnels regardless of random seeds, it is possible that all basins obtained by SGD are fully path-connected.

---

### Author Response · Authors · 2025-12-01
**A summary of rebuttal (part 1)**

We thank the Area Chair and all reviewers for their evaluation of our work and for the constructive feedback. The rebuttal phase has given us a valuable opportunity to improve our work, and we are especially grateful to Reviewers WDuq and suKz for their detailed reading, insightful questions, and suggestions that directly informed new experiments. These new experiments test our algorithm in more extreme situations and find that our algorithm still works, which broadens its applicability.

At the same time, we observed that two reviewers appear to write comments based on misunderstandings of our work.
Reviewer WGfT seems to conflate our work with linear mode connectivity, whereas our paper is about connecting independently trained models via non-linear low-loss paths.
Reviewer 9MCz characterizes our algorithm as “trivial” and suggests that finding a low-loss path between independently trained modes is not a meaningful challenge.
Given that this topic is sparsely explored and to the best of our knowledge, fewer than three empirical algorithms capable of finding such low-loss paths, we believe these comments underestimate the novelty and difficulty of the setting. We respectfully ask the Area Chair to consider these clarifications when interpreting the reviews and scores.


In response to the reviews, we have made the following changes and additions to improve our paper:

1. Revise the motivation paragraph.

We revise the first paragraph to clarify our motivation is to come up with a algorithm framework to navagate in the low-loss space, which further allows geometric investigation of low-loss space.

2. Conduct addtional experiments to connect modes in sharp minima to verify our algorihtm is roubust for sharp minima.

In the original submission, we hypothesized that starting from sharp minima might cause our algorithm to fail. Following Reviewer WDuq’s suggestion, we conducted targeted experiments on sharp minima with near-zero training loss, obtained via unflavorable training hyperparameters. Contrary to our initial concern, our algorithm consistently finds feasible low-loss paths between such modes. These new results are reported in Appendix “Connecting modes located in sharper minima”.

3. Verify our algorihtm could empirically work for all SGD solutions.

Reviewer suKz raised concerns about the applicability range of our proposed method.
We provide a systematic empirical study showing that our algorihtm could work for all SGD solutions up to the point where training fails to converge. These experiments and analyses are included in Appendix “Empirical investigation of the variance sphere range supported by Algorithm 2”.

3. Add a Discussion section to discuss our empirical results could reveal deeper structural properties of loss landscape.

As Reviewer WDuq noted, our work suggests deeper structural properties of the loss landscape. Due to page limits, we had originally removed this discussion. We have now reinstated and expanded it in a dedicated Discussion section. Briefly, our empirical results indicate that, for the models and datasets covered in our paper, low-loss solutions obtained by SGD appear to lie on a single, fully path-connected low-loss manifold: independently trained modes can always be connected by continuous low-loss paths without leaving this region.

4. Additional improvements.

- We revise the first paragraph in the Introduction section to clarify the motivation.
- We add AutoNEB results and runtime comparisons, and directly compare them with our method.
- We conducted additional experiments on CIFAR-100. The corresponding results are reported in Appendix 3 and Appendix 4.

---

> ### Author Response · Authors · 2025-12-01
> **Summary of rebuttal (part 2)**
>
> Meanwhile, we are afraid Reviewer WGfT and Reviewer 9MCz do not fully understand our work, which raises misunderstanding in our work and linear mode connectivity, finally lead to low scores. We present the evidence below:
>
> ***Reviewer WGfT:***
>
> 1. Reviewer WGfT recommends citing [Juneja 2022] and [Altıntaş 2025] to discuss transformer-based models in NLP. However, both of these works study Linear Mode Connectivity (LMC), whereas our paper focuses on mode connectivity between independently trained modes, a setting not addressed by LMC. We explicitly clarify this distinction in the section “Mode Connectivity in Spawning and Permutation”(initial submission): our goal is to connect independently trained solutions.
>
> 2. While recommending LMC papers, Reviewer WGfT also describes our work as “revisiting non-linear mode connectivity” in Strength bullet point 1. This suggests that the reviewer is aware that our work is about non-linear mode connectivity, yet several later comments treat our setting as if it were linear mode connectivity, leading to an internal inconsistency in their assessment.
>
> 3. Reviewer WGfT also overlooks key information already presented in the paper. Reviewer WGfT suggests that we should investigate transformer models (Weakness 1 and 2). However, we already evaluate Compact Convolutional Transformers (CCT), as stated in the abstract, and present CCT as one of our main results in Figure 2 and Figure 3 (initial submission). Thus, the requested experiment is in fact already included.
>
> 4. Reviewer WGfT notes in Weakness bullet point 6: "The benefits of the algorithm is not validated. According to Table 6, algorithm 1 takes 21 hours on an H100 GPU for a ResNet18 on Cifar-10. If I am not misreading this table, the cost of the algorithm is quite high. Training a ResNet18 or 20 on Cifar-10 shouldn’t take more than 10 minutes." Our objective, as clearly stated in the title “Connecting Independently Trained Modes via Layer-Wise Connectivity”, is to find a low-loss path connecting two independently trained modes, not merely to train a single model. A path consists of a sequence of intermediate models, each of which must remain low-loss. It is therefore expected that the computational cost of constructing such a path is substantially higher than that of training a single point solution. Comparing the runtime of path-finding to the runtime of training one model conflates these fundamentally different tasks.
>
> Taking points 2 and 4 together, one possible explanation is that Reviewer WGfT relied on an LLM tool to initially read and digest our paper: the LLM correctly summarized that our work revisits non-linear mode connectivity, but the reviewer later seems to revert to interpreting it within the linear mode connectivity setting. This would also naturally explain the comment that our algorithm is “slow,” since constructing a simple linear interpolation is of course much cheaper than finding a non-linear low-loss path between independently trained modes.
>
>
>
> ***Reviewer 9MCz:***
>
> Reviewer 9MCz provides only two brief comments, both of which appear to overlook central parts of the paper.
>
> 1. Treating mode connectivity as a “given” phenomenon
>
> Reviewer 9MCz writes:
> “In my understanding, mode connectivity is more like a phenomenon that helps us better understand the loss landscape, instead of a challenge that needs to be solved by an algorithm.”
> To the best of our knowledge, there is currently no work showing that mode connectivity is a universal phenomenon across architectures, nor a clear criterion for when it should hold. As we state in the initial submission:
> “The primary limitation is the narrow range of model architectures supported by existing methods. Simple architectures require only simple tools … leaving an open question of whether mode connectivity persists in more recent and sophisticated models.”
> Our work precisely aims to extend mode connectivity to more recent and complex architectures.
>
> 2. Overlooking explanations already provided for key algorithmic steps.
>
> Reviewer 9MCz also overlooks information in our paper. For example, the reviewer comments that “The algorithm looks trivial, and is not explained at all. For example, why do you need to project back to the variance sphere at each step?”
>
> However, the answer to this question is right in the original text (second paragraph of “Low-Loss Path Finding Algorithm” section): "Next, the VarianceCorrection step projects $M_1$ back as $M_2$ onto the variance sphere of $P_0$ to counteract the vanishing variance problem, a phenomenon wherein averaging uncorrelated neural networks leads to reduced parameter variance, thereby hindering subsequent training efforts(Tian et al., 2024)."
>
> In addition, we also do not think one can say an algorithm is trivial if he does not understand it.

---

### Meta-Review · Area_Chair_MMvx · 2026-01-04

**Summary:**

The paper aims to extend the mode connectivity phenomenon [1] to more modern architectures and more diverse evaluation settings. To do so, the authors propose a new method for connecting independently trained models, and demonstrate that it works with newer architectures not considered in [1]. The authors also show that the method works in connecting models trained with different hyper-parameters.

I believe, there is a major issue with the paper in its current form. Specifically, the logic is as follows.
- [1] and others demonstrated mode connectivity on ResNets, VGG, Wide ResNets and smaller CNNs; they did not consider EfficientNet, ShuffleNet, RegNet, DLA and CCT.
- The authors develop a new method for mode connectivity and show that it works for these newer architectures.
- **The authors do not check whether existing methods would work for these architectures** or for connecting models trained with different hyper-parameters.

Moreover, in the rebuttal period the authors explicitly state:
> If the concern is whether prior methods might also work on these architectures, we agree this is theoretically possible. However, in past six years, no prior work has demonstrated such results

While this argument suggests that there is merit in demonstrating mode connectivity on these architectures, it is not a valid justification for the proposed method. If the existing methods work on these architectures, the proposed method is unnecessary.

Moreover, the authors make factually incorrect claims in the paper and the rebuttal about the work [1].
> The concept of mode connectivity was first introduced by Garipov et al. (2018), who proposed Fast Geometric Ensembling (FGE) to generate an ensemble model from two modes. Their method does not explicitly construct a mode connection, but rather states its existence.

The paper [1] in fact proposes two methods: the first "Connection procedure", section 3.1 is a method for explicitly constructing paths between optima. The second is the FGE, which is a separate ensembling method. Importantly, the curve-finding method of [1] constructs a mode-connecting path by effectively training one additional model in time comparable to standard training. On the other hand, the method proposed by the authors is highly complex, with multiple hyper-parameters, and
> In the ResNet-18@CIFAR-10 example (Figure 2), the low-loss path consists of 35,000 intermediate low-loss models.

In the case of ResNet-18, training one model takes approximately 10 minutes, while running the mode-connecting procedure takes 21 hours.

To sum up,
- The authors misrepresent prior work.
- The proposed method is dramatically more expensive than methods developed in the prior work.
- The authors do not show that the method is capable of solving problems that prior methods could not solve.

It is likely that the proposed method finds mode-connecting paths with smaller barriers than prior work, but at a dramatic cost. It is unclear what the value of this improvement is.

Based on the above, I believe in its current form the main contribution of the paper is demonstrating mode connectivity on an extended collection of architectures. However, the paper basically follows the 2018 paper [1] in terms of datasets, and the current evaluation is not sufficient to claim that it demonstrates mode connectivity in modern practical settings.

[1] _Loss Surfaces, Mode Connectivity, and Fast Ensembling of DNNs_
Garipov, Izmailov, Podoprikhin, Vetrov, Wilson;
Neural Information Processing Systems (NeurIPS), 2018

**Reviewer Concerns:**

- The authors added experiments connecting models with more different hyperparameters (addressed)
- Minor presentation issues (addressed)
- Experiments related to connecting sharp and flat optima (experiments added, addressed)
- Experiments on small scale datasets only (not addressed; added CIFAR-100, but that's the same scale)
- Concerns about the details of the method (partially addressed, authors explained motivation)
- Concerns around motivation for this work and its potential impact (not addressed)
- Concerns about the cost of the method (not addressed)
- Concerns about comparison to existing methods (not addressed)

**Reviewer Scores:**

- suKz: 4 -> 4; it is possible the reviewer would raise to a 6, but concerns about scale of experiments not addressed
- 9MCz: 2 -> 2; concerns about the motivation for the method not addressed
- WDuq: 6 -> 6; did not have major concerns, mostly suggestions and questions around flatness which the authors discussed adequately
- WGfT: 2 -> 2; concerns about comparison to baselines, motivation and scale of the experiments not addressed

---

### Decision · Program_Chairs · 2026-01-26

Reject